



# Quantifying riming from airborne data during HALO-(AC)[3]

Nina Maherndl[1], Manuel Moser[2,3], Johannes Lucke[3,4], Mario Mech[5], Nils Risse[5], Imke Schirmacher[5], and Maximilian Maahn[1]

[1]Leipzig Institute of Meteorology (LIM), Leipzig University, Leipzig, Germany
[2]Institute for Physics of the Atmosphere, Johannes Gutenberg University, Mainz, Germany
[3]Institute for Physics of the Atmosphere, German Aerospace Center (DLR), Wessling, Germany
[4]Faculty of Aerospace Engineering, Delft University of Technology, Delft 2629, the Netherlands
[5]Institute for Geophysics and Meteorology, University of Cologne, Cologne, Germany

**Correspondence:** Nina Maherndl (nina.maherndl@uni-leipzig.de)

**Abstract.** Riming is a key precipitation formation process in mixed-phase clouds by efficiently converting cloud liquid to ice water. Here, we present two methods to quantify riming of ice particles from airborne observations with the normalized rime mass, which is the ratio of rime mass to the mass of a size-equivalent spherical graupel particle. We use data obtained during the HALO-(AC)³ aircraft campaign, where two aircraft were collecting spatially and temporally closely collocated radar and

in situ measurements over the Fram Strait west of Svalbard in spring 2022. The first method is based on an inverse Optimal Estimation algorithm to retrieve the normalized rime mass from a closure between cloud radar and in situ measurements during these collocated flight segments ("combined method"). The second method relies on in situ observations only, by relating the normalized rime mass to optical particle shape measurements ("in situ method"). We find good agreement between both methods during collocated flight segments with median normalized rime masses of 0.018 and 0.016 (mean values of 0.027

and 0.028) for combined and in situ method, respectively. Assuming particles with a normalized rime mass smaller 0.01 to be unrimed, we obtain average rimed fractions of 77 % and 75 %. Although in situ measurement volumes are in the range of a few $cm^3$ and therefore much smaller than the radar volume (about 45 m footprint diameter), we assume they are representative of the radar volume. When this assumption is not met due to less homogeneous conditions, discrepancies between the two methods result. We compare normalized rime mass results with meteorological and cloud parameters and show the performance of the

methods in two case studies, 1) a collocated segment in cold air outbreak conditions and 2) an in situ only flight close to a polar low. We find that higher normalized rime masses correlate with streaks of higher radar reflectivity. We also observe rimed particles in regions without liquid water, suggesting that particles were rimed in a liquid layer above and precipitated. The methods presented improve our ability to quantify riming from aircraft observations.

## 1  Introduction

Mixed-phase clouds (MPC) are a crucial part of the Arctic climate system. Observations have shown that MPC occur about 40 % of the time (e.g., at Barrow, Alaska or Ny-Alesund, Svalbard; Shupe, 2011; Gierens et al., 2020), can persist up to several days (Zuidema et al., 2005), and can span hundreds of kilometers by forming organized cloud streets during cold air outbreaks (Müller et al., 1999). MPC play a critical role in the Arctic hydrological cycle and radiation budget, having on average a positive



surface radiative forcing (Shupe and Intrieri, 2004; Kay and L'Ecuyer, 2013). However, the role of MPC in a rapidly warming
Arctic ("Arctic Amplification"), where the mean near-surface air temperature has increased nearly four times more than the
global mean during the last four decades (Rantanen et al., 2022), is not fully understood yet. It is unclear, whether changes in
MPC properties or frequency of occurrence will accelerate or decelerate Arctic amplification (Wendisch et al., 2023).

MPC properties are in part determined by microphysical processes. Supercooled liquid water droplets (SLW) can coexist
with ice particles in MPC between $0\,°C$ and about $-38\,°C$; at colder temperatures, homogeneous freezing occurs. Typically,
MPC are composed of a single or multiple stratiform layers of SLW near the cloud top and ice particles within and beneath the
SLW layers (Shupe et al., 2006). While this composition is thermodynamically unstable, long MPC life times are driven by
a combination of various processes and feedback mechanisms that are poorly understood (Morrison et al., 2012). The repre-
sentation of these processes poses a major source of uncertainty in numerical weather forecast and climate models (Morrison
et al., 2020).

One important ice growth process, besides aggregation and depositional growth, common in MPC is riming. Riming occurs,
when SLW comes into contact with ice particles, freezing onto them almost instantly. Typically, riming leads to denser, more
spherical ice particles with increased mass, size, and fall velocity (Heymsfield, 1982; Erfani and Mitchell, 2017; Seifert et al.,
2019). Due to its efficiency in converting SLW, riming is a key process for ice growth and subsequent precipitation formation.
Moisseev et al. (2017) showed that in Hyytiälä (Finland) riming was responsible for $5\,\%$ to $40\,\%$ of snowfall mass during
winter 2014/2015 whereas Harimaya and Sato (1989) found riming proportions above $50\,\%$ for snowfall in a Japanese seaside
area in 1987. Nonetheless, riming is often neglected in studies of Arctic MPC (Avramov et al., 2011; Yang et al., 2013; Oue
et al., 2016), especially in cases with low liquid water paths (LWP). Fitch and Garrett (2022) showed in a recent study that
riming is very common in Arctic low level MPC, also in cases of LWP less than $50\,\mathrm{g\,m^{-2}}$. Only $34\,\%$ of precipitating particles
observed at Oliktok Point, Alaska showed negligible amounts of riming. They proposed that riming enhancement can occur in
regions with updrafts so that particles are exposed to SLW for a longer time span before falling out.

Riming has been studied in situ by airborne or ground-based measurements. Individual ice crystals or snowflakes that are
observed manually (Harimaya and Sato, 1989; Mosimann et al., 1994) or by optical probes (Praz et al., 2017; Waitz et al.,
2022) are often qualitatively classified. Mosimann (1995) was the first to quantify the degree of snow crystal riming using
radar Doppler velocity measurements. They defined the riming degree on a scale from 0 to 5, where 0 means unrimed, 3 means
heavily rimed and 5 means graupel. Mason et al. (2018) retrieved a "density factor" as a proxy for riming from dual-frequency
radar Doppler velocity measurements. Kneifel and Moisseev (2020) presented long-term statistics of the rime mass fraction
(FR), the ratio of rime mass and snow particle mass, also obtained by Doppler velocity measurements, whereas Vogl et al.
(2022) showed that FR can be predicted by an artificial neural network from radar reflectivity $Z_e$ and skewness measurements.
Previous studies have shown that collocating radar signals and in situ cloud data can be used to create, improve and validate
microphysical cloud retrievals (Tian et al., 2016; Trömel et al., 2021; Blanke et al., 2023).

In the Arctic, there are only few observations of riming given the difficulty of 1) obtaining (quantitative) measurements of
riming in general and 2) performing cloud measurements in remote regions. Airborne campaigns offer unique opportunities
of measuring in regions that are otherwise inaccessible. Waitz et al. (2022) showed observations of ice particles by optical



probes collected during ACLOUD (Arctic CLoud Observations Using airborne measurements during polar Day, May/June
2017 based in Svalbard, Wendisch et al., 2019). Images of ice particles are observed manually and qualitatively classified as
"unrimed", "slightly rimed", "moderately rimed", "heavily rimed", and "graupel". Nguyen et al. (2022) presented coincident
triple frequency radar and in situ observation obtained during the RadSnowExp (Radiation Snow Experiment, fall 2018 based
in Iqaluit, Canada; Wolde et al., 2019). They show close relationships between the triple-frequency signatures and in situ
derived effective ice particle bulk density, which functions as a proxy for riming. Further, they compare to a machine learning
ice particle habit classification that includes "rimed" categories. While both Waitz et al. (2022) and Nguyen et al. (2022) show
the common occurrence of riming in Arctic MPC and the high value of aircraft observation to study riming, neither method
can quantify the fraction riming contributes to particles' mass.

In this study, we present two methods to quantify riming from airborne measurements and apply them to data collected during
the aircraft campaign Arctic Air Mass Transformations during Warm Air Intrusions and Marine Cold Air Outbreaks (HALO-
$(AC)^3$). HALO-$(AC)^3$ took place in March/April 2022 with the main objective to study Arctic air mass transformations and
conduct collocated measurements with up to three aircraft. We focus on (collocated) remote sensing and in situ measurements
obtained with the research aircraft Polar 5 and Polar 6, respectively. Both aircraft were based in Svalbard and measurements
were mainly collected over the open ocean and in the marginal sea ice zone (MIZ), the transition zone between open ocean and
closed sea ice, west of Svalbard. We use the normalized rime mass $M$ (Seifert et al., 2019), the ratio of rime mass to the mass
of an equally large graupel particle, to quantify riming. The first method is based on an Optimal Estimation algorithm to obtain
$M$ from a closure between cloud radar and in situ measurements during collocated flight segments ("combined method", see
Sect. 3.1). We find $M$ by matching measured radar reflectivities $Z_e$ with simulated $Z_e$ obtained from observed in situ particle
number concentrations. The second method derives $M$ from in situ measured particle shape ("in situ method", see Sect. 3.2).
We compare results for $M$ obtained with both methods for all collocated flight segments and investigate their relation to
meteorological and cloud parameters such as temperature, liquid water content (LWC), total water content (TWC), LWP, and
in-cloud location (Sect. 4.1). Further, we analyze all in situ data and evaluate how representative the collocated segments are
for the whole campaign (Sect. 4.2). We then show two case studies: 1) a collocated flight segment from 01 April during cold
air outbreak conditions (Sect. 4.3) and 2) a flight segment from 08 April in the proximity of a polar low (Sect. 4.4).

## 2 Data

### 2.1 The HALO-$(AC)^3$ airborne campaign

In this study, radar and in situ data from the HALO-$(AC)^3$ campaign (Wendisch, M. et al., 2023) are analyzed. During the
campaign organized by the Transregional Collaborative Research Centre TR 172 $(AC)^3$, three research aircraft were employed
to study the Arctic atmosphere. The main objectives of the campaign included investigating warm air intrusions into the Arctic
as well as marine cold air outbreaks (MCAO) and collecting collocated measurements with up to three aircraft (Wendisch, M.
et al., 2023). The synoptic situation during the campaign is described in Walbröl et al. (2023). The instrumentation on board
the Polar aircraft is similar to the one used during the Airborne measurements of radiative and turbulent FLUXes of energy



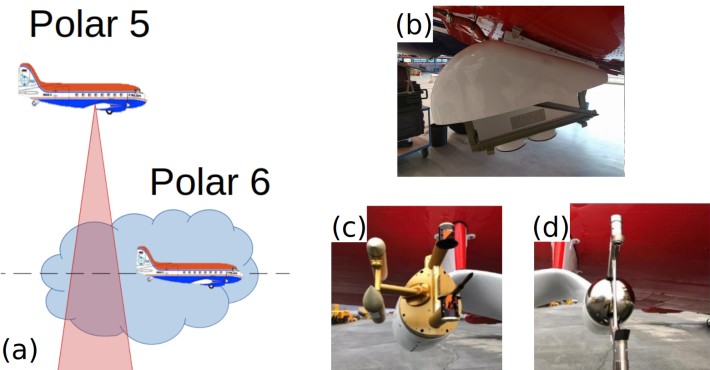

**Figure 1.** (a) Concept of collocation: while a radar onboard Polar 5 is measuring the cloud from above, cloud probes onboard Polar 6 simultaneously collect in situ samples at (almost) the same location inside the cloud. (b) MiRAC-A on Polar 5 in its belly pot and the wing-mounted cloud probes (c) Cloud Droplet Probe (CDP), Cloud Imaging Probe (CIP) and (d) Precipitation Imaging Probe (PIP) on Polar 6.

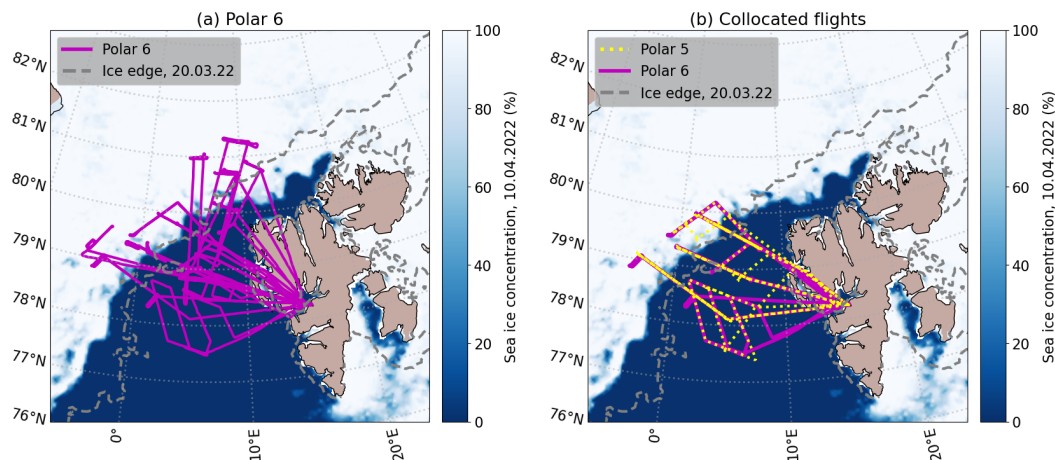

**Figure 2.** Flight tracks of (a) all Polar 6 flights (in situ) conducted during HALO-(AC)[3] and (b) flights with collocated Polar 5 (remote sensing) and Polar 6 segments. The sea ice concentration (SIC) derived from the Advanced Microwave Scanning Radiometer 2 (AMSR2) onboard the GCOM-W1 satellite on 10 April (at campaign end) is shaded in blue; The ice edge (15 % SIC) on 20 March (at campaign start) is shown in light gray.

and momentum in the Arctic boundary layer (AFLUX) and Multidisciplinary drifting Observatory for the Study of Arctic Climate – Airborne observations in the Central Arctic (MOSAiC-ACA) campaigns described in Mech et al. (2022a). During the majority of flights analyzed in this study, north and northeasterly wind transported cold air masses from the central Arctic to the main measurement area in the Fram Strait.

We focus on data collected by Polar 5 and Polar 6, two Basler BT-67 aircraft operated by the Alfred Wegener Institute, Helmholtz Centre for Polar and Marine Research (AWI; Wesche et al., 2016). A total of 11 flights with Polar 5 and 13 with



Polar 6 were conducted in March and April 2022 during HALO-(AC)[3] in the vicinity of Svalbard. Closely collocated and nearly coincident measurements were obtained with the W-band cloud radar component of the Microwave Radar/radiometer for Arctic Clouds (MiRAC-A, Mech et al., 2019) on board of Polar 5 and a variety of in situ cloud probes mounted under the wings of Polar 6 (Mech et al., 2022a). Figure 1 (a) shows a conceptual sketch of how collocation was achieved: while Polar 6 was flying low and in-cloud, Polar 5 was following in close proximity on the same track above. The slight off-set between the two planes was necessary so that dropsondes could be released safely from Polar 5.

Figure 2 shows (a) all flight tracks of Polar 6 and (b) flight tracks of both aircraft for flights with collocated segments. The overlapping lines show close spatial collocation. The sea ice concentration (SIC) at campaign beginning and end indicate the variable sea ice conditions. All 13 Polar 6 flights result in over 60 hours of flight time and about 32 hours of cloud particle measurements. 31 % of the total flight time during the flights shown in Fig. 2 (b) was conducted "collocated", which we define as both aircraft having a maximum horizontal distance of 5 km within a 5 min time window. From a total of about 11.8 hours of collocated flight time, 4.6 hours are collocated cloud measurements (this corresponds to a distance of approximately 1300 km assuming a typical speed of 80 $\mathrm{ms^{-1}}$). The analyzed data covers a temperature range of -31 to -1 °C and an altitude range of in cloud measurements from close to the ground to 1760 m.

## 2.2 In situ cloud probes

During HALO-(AC)[3], a variety of in situ cloud data was collected. This study uses microphysical cloud data collected from three different cloud instruments, the Cloud Droplet Probe (CDP; Lance et al., 2010; Wendisch et al., 1996), the Cloud Imaging Probe (CIP; Baumgardner et al., 2011) and the Precipitation Imaging Probe (PIP; Baumgardner et al., 2011). All three probes were installed under the wings of Polar 6 (Fig. 1) and operated by the German Aerospace Center (DLR). The CDP is a forward-scattering optical spectrometer. The instrument measures cloud particles in the size range 2.8 to 50 µm by the intensity of forward scattered laser light underlying Mie theory. Larger cloud particles are measured via Optical Array Probes (OAP). Here, two-dimensional shadow images of the cloud particles are recorded as the particles pass through the instrument's sampling area. The collected data by the CIP and PIP differ in pixel resolution. Both instruments consist of a 64 diode array with the CIP covering a size range from 15 µm to 960 µm (15 µm resolution) and the PIP covering from 103 µm to 6.4 mm (103 µm resolution). By combining CDP, CIP and PIP, a continuous particle size distribution is derived including all hydrometeors from 2.8 to 6400 µm. The same processing methods of the OAP data is applied as used for the AFLUX and MOSAiC-ACA campaings (Moser and Voigt, 2022; Moser et al., 2022). The operating principle of the respective instruments, processing, uncertainties and applied corrections is described in detail by Moser et al. (2023) and Mech et al. (2022a).

Liquid water content (LWC) and total water content (TWC) were measured with a Nevzorov probe (Korolev et al., 1998). The probe was operated with a new sensor head, which featured an LWC sensor and two TWC cones with diameters of 8 and 12 mm (Lucke et al., 2022). The Nevzorov probe contains sensing elements which are regulated to provide a constant temperature, (110°C during the HALO-(AC)[3] campaign). Droplets and ice particles momentarily cool the sensing elements when they impinge. In consequence, the sensors draw more power as they heat and evaporate impinging water in order to maintain their temperature, which can be used to estimate bulk LWC and bulk TWC. The measurement range of the Nevzorov





probe extends from approximately 0.01 to 3.0 $\mathrm{gm}^{-3}$. Uncertainties of the Nevzorov depend very much on the atmospheric conditions that are present (Lucke et al., 2022). Nevzorov probe measurements during HALO-(AC)[3] are only available for flights in April due to technical difficulties in March. Air temperature was measured with a Pt100 mounted in a Rosemount-
housing at the noseboom of Polar 6. The measurements were corrected for adiabatic heating in the housing.

With the collected data, we are unable to distinguish between larger liquid droplets and small solid ice particles due to low resolution images consisting of only a few pixels. Similar to Moser et al. (2023), we assume all particles larger (smaller) 50 µm to be ice crystals (liquid droplets), which is an appropriate assumption for the majority of low level Arctic MPC (McFarquhar et al., 2007; Korolev et al., 2017). This assumption is based on the good agreement between Nevzorov probe LWC and LWC
calculated from the PSD assuming particles smaller 50 µm to be liquid droplets where both measurements are available ($R^2$ = 0.83; Nevzorov and PSD LWC sum up to 973 and 983 $\mathrm{gm}^{-3}$, respectively, and lie within 1 % of each other). Additionally, we do not expect this assumption will lead to significant biases due to radar reflectivities (that we simulate from in situ PSDs) being dominated by large particles.

## 2.3  Airborne remote sensing instruments

The Microwave Radar/radiometer for Arctic Clouds (MiRAC; Mech et al., 2019) was designed for operation on board the research aircraft Polar 5. During HALO-(AC)[3] the active radar component (MiRAC-A) was operated on board of Polar 5 in the same constellation as during MOSAiC-ACA. MiRAC-A is a 94 GHz frequency-modulated continuous wave (FMCW) radar, which was mounted with an inclination angle of 25° backward in a belly pod under Polar 5. The radar measurements have been quality controlled and corrected for surface clutter, mounting of the instrument, and aircraft attitude (Mech et al., 2019).
This results in geo-referenced, regularly gridded data with a vertical resolution of 5 m (with reliable measurements starting 150 m above ground level due to ground clutter effects and 200 m distance from the aircraft for full overlap). Mech et al. (2019) estimate the accuracy of the radar reflectivity $Z_e$ calibration to be 0.5 dBZ (neglecting attenuation). Because Doppler velocity measurements are biased by the aircraft motion, only $Z_e$ measurements are used in this study.

The MiRAC-A radar is also equipped with a horizontally-polarized 89 GHz passive channel using the same antenna as the
radar. The brightness temperature ($T_B$) is also measured under a tilted angle of 25° backwards to nadir. From this observations, the liquid water path (LWP) is estimated over open ocean only with a temporal resolution of 1 s as described in Ruiz-Donoso et al. (2020). Thereby, the retrieval for the LWP is based on $T_B$ derived from simulations with the Passive and Active Microwave radiative TRAnsfer tool (PAMTRA, Mech et al., 2020) using profiles of nearby dropsondes and artificial LWPs as input. To eliminate biases in the observations, the difference between clear-sky and cloudy observations were used. Due to the variable
microwave emissivity of sea ice, the LWP product is only available above open ocean.

Cloud top height (CTH) is obtained from the Airborne Mobile Aerosol Lidar for Arctic research (AMALi; Stachlewska et al., 2010), which was also operated on Polar 5. AMALi measures backscatter intensity profiles at 532 nm (polarized) and 355 nm (not polarized), from which the attenuated backscatter coefficient is calculated (Ehrlich et al., 2019). CTH is determined by searching for gradients in the backscatter coefficient.



For the present study, $Z_e$ has been corrected for attenuation due to atmospheric gases as well as liquid hydrometeors. The two-way attenuation profile was calculated with PAMTRA. We used measurements from the closest dropsonde and the water vapor absorption model by Rosenkranz (Rosenkranz, 1998) to calculate the attenuation due to water vapor for each time step. To estimate attenuation due to liquid water, we took LWC measurements from the Nevzorov probe operated onboard Polar 6 during the closest vertical cloud profile. Whenever Nevzorov probe measurements were not available, LWC was calculated by

integrating the particle size distribution (PSD) of liquid particles ($< 50\,\mu m$) measured with the cloud probes on board Polar 6. In both cases, LWC measurements were averaged to be on a regular vertical grid with a resolution of $10\,m$. Attenuation due to snowfall is assumed to be negligible compared to liquid droplets. During HALO-(AC)³ we obtain a mean two-way attenuation of $0.41\,dB$. By comparing integrated LWC measured with the Nevzorov probe and integrated LWC calculated from PSD during cloud profiles (if both are available) to the temporally higher resolved LWP from MiRAC-A, we estimate uncertainties of the

attenuation correction to be $1\,dB$ leading to a total uncertainty of $Z_e$ of $1.5\,dB$.

## 2.4  Collocation of radar and in situ measurements

In order to combine radar and in situ measurements, it is critical to have a temporally and spatially collocated data set. Following Chase et al. (2018) and Nguyen et al. (2022), the nearest radar data point to the in situ measurements is selected. We matched each 1 Hz Polar 5 data point with the spatially closest Polar 6 data point with a maximum horizontal distance of $5\,km$ within a 5

min time window. Further, the radar range gate closest to the flight altitude of Polar 6 was chosen. Averaging radar reflectivity over certain height ranges close to Polar 6 did not lead to improvements. A rolling average of $30\,s$ was applied to in situ data to obtain more robust statistics and to the radar data to make results comparable. Also, this is done to compensate for the different sampling volumes to a certain extent. While the radar footprint of a cloud in $2500\,m$ distance is approximately $45.15\,m$ in diameter, the cloud probes have measurement volumes in the range of a few cubic centimeters. We are aware that the

assumption that the in situ measurement is representative of the entire matched radar volume is not always met and discuss possible implications of the assumption to our results in Sect. 5.

## 2.5  Simulated rimed aggregates

In addition to the observations, we use a data set of simulated rimed aggregates to relate particle properties and riming as discussed in Maherndl et al. (2023a). The aggregation and riming model described in Leinonen et al. (2013), Leinonen and

Szyrmer (2015) and Leinonen and Moisseev (2015) is used in the setting "B" (aggregation followed by riming) to generate aggregates built from a predefined number of monomer crystals. The monomer crystal sizes are taken from an exponential size distribution and the crystals themselves are composed of cubic volume elements with an edge length of $20\,\mu m$. The aggregate sizes range from slightly below $100\,\mu m$ to $12\,mm$. In this study, we only use dendrite monomer crystals, which is motivated by manual inspection of the in situ images for the collocated flight segments. After aggregation, the particles are exposed to

a predefined amount of liquid water so that riming occurs. The frozen droplets that have rimed onto the ice particles are also represented by cubic volume elements of $20\,\mu m$.



## 3 Methodology

Here, we describe how we obtain quantitative measures of riming in two different ways. To quantify riming, we use the normalized rime mass $M$ (Seifert et al., 2019), which is defined as the rime mass $m_{\mathrm{rime}}$ divided by the mass of the size-equivalent spherical graupel particle $m_g$, where we assume a rime density of $\rho_{rime} = 700 \text{ kg m}^{-3}$:

$$M = \frac{m_{\mathrm{rime}}}{m_g} \tag{1}$$

where

$$m_g = \frac{\pi}{6} \rho_{\mathrm{rime}} D_{\mathrm{max}}^3. \tag{2}$$

The definition of $M$ implies $M = 0$ for unrimed particles and $M \to 1$ for heavily rimed, spherical graupel particles. The maximum dimension $D_{\mathrm{max}}$ is defined as the diameter of the smallest encompassing circle and is used to parameterize particle sizes during the whole study.

First, we present an algorithm based on Optimal Estimation (Rodgers, 2000; Maahn et al., 2020) to retrieve average $M$ of observed cloud particle populations for each time step from a closure of collocated remote sensing and in situ data ("combined method"; Sect. 3.1). Second, we describe the calculation of $M$ based on in situ measured cloud particle shape. We calculate the complexity $\chi$ from in situ images and use a data set of simulated rimed aggregates to relate $\chi$ to $M$ ("in situ method"; Sect. 3.2).

### 3.1 Combined method

We take advantage of collocated Polar 5 and Polar 6 flights and retrieve the average $M$ of the observed cloud particle population value for each time step from the combination of radar and in situ measurements (Fig. 3).

First, $Z_e$ is corrected for attenuation (see Sect. 2.3), all cloud edges are removed to avoid non uniform beam filling and Polar 5 and Polar 6 data is combined (see Sect. 2.4). PAMTRA (Mech et al., 2020) is used to simulate $Z_e$ from the in situ PSD and an initial guess for $M$. In principle, $Z_e$ is a function of the mass, PSD, and scattering properties of the observed particle population. If the PSD is known and particle mass and backscattering are parameterized as a function of riming, $M$ can be derived from a closure of $Z_e$ and PSD. In the retrieval radar forward operator, we use Mie scattering (Mie, 1908) for liquid droplets. For ice particles, we use the Self-Similar Rayleigh-Gans approximation (SSRGA; Hogan and Westbrook, 2014; Hogan et al., 2017) and calculate the required SSRGA parameters from $M$ with the empirical relations presented in Maherndl et al. (2023a). In addition, we consider the mass-size relation to follow a power law ($m = a_m \cdot D_{\mathrm{max}}^{b_m}$), take the mass-size parameters $a_m$ and $b_m$ for dendrites from the same study and interpolate $a_m$ and $b_m$ to obtain parameters for a continuous $M$.

By parameterizing scattering as well as mass-size relations only by $M$ and assuming that the measured PSDs are representative of radar measurements, we can tweak $M$ until measured and forward simulated radar reflectivities match within a given uncertainty range. This is done by Optimal Estimation (OE), a retrieval technique based on Bayes' theorem (Rodgers, 2000) implemented in pyOptimalEstimation (Maahn et al., 2020). OE uses a priori information $\mathbf{x}_a$ and a Gaussian statistical





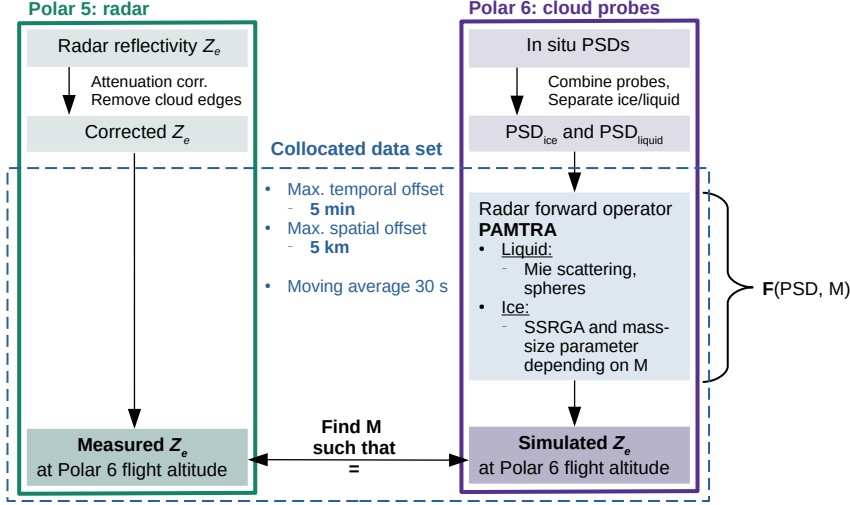

**Figure 3.** Schematic of the retrieval framework.

model to estimate the state vector $\mathbf{x}$ from the observation vector $\mathbf{y}$ in an iterative scheme. Starting with $\mathbf{x}_a$ as a first guess for $\mathbf{x}$, the forward model $F(\mathbf{x})$ (i.e., PAMTRA) is used to convert state to observation space. Then, the difference between $\mathbf{y}$ and

$F(\mathbf{x}_a)$ is used to make a next guess for the state vector $\mathbf{x}_1$, which requires inverting $F(\mathbf{x})$ with the help of the Jacobian matrix $\mathbf{K} = \partial F(\mathbf{x})/\partial \mathbf{x}$. This scheme is repeated until the posteriori probability distribution $P(\mathbf{x}|\mathbf{y}) = P(\mathbf{y}|\mathbf{x})/P(\mathbf{y})$ reaches a maximum resulting in the optimal $\mathbf{x}$. This is achieved by minimizing the cost function $J$:

$$J = [\mathbf{y} - F(\mathbf{x})]^T \mathbf{S_y}^{-1}[\mathbf{y} - F(\mathbf{x})] + (\mathbf{x} - \mathbf{x_a})^T \mathbf{S_a}^{-1}(\mathbf{x} - \mathbf{x_a}) \tag{3}$$

where $\mathbf{S}_y$ is the uncertainty of $\mathbf{y}$ (observation covariance matrix) and $\mathbf{S}_a$ the a-priori uncertainty (covariance matrix of $\mathbf{x}_a$).

Given that our problem is unambigious (one measurement parameter $\mathbf{y}$ ($Z_e$) and one state parameter $\mathbf{x}$ ($M$)), using OE is not strictly necessary but has the advantage of providing uncertainties.

We chose $\mathbf{x}$ to represent $M$ in common logarithmic scale to avoid negative values and make $\mathbf{S}_a$ more Gaussian. We use $\mathbf{x}_a = -1$ (corresponding to $M = 0.1$) as a-priori information and $\mathbf{S}_a = 1$ as a-priori uncertainty. We also evaluated different a-priori guesses $\mathbf{x}_a$ and uncertainties $\mathbf{S}_a$, but they lead to almost identical results and are therefore not shown. $\mathbf{y}$ are the attenuation

corrected $Z_e$ measurements in Polar 6 flight altitude and $\mathbf{S}_y$ is the corresponding measurement uncertainty of $1.5$ dB. In Appendix A we show that the OE output captures uncertainties of the combined method with synthetic data.



## 3.2 In situ method

The second method exploits the fact that riming impacts ice particle shape and typically leads to more spherical particles that can be derived from in situ image properties obtained by the CIP and PIP. This method can, in principle, be applied to all Polar

6 cloud particle measurements. From the captured images, hydrometeor properties described in the following were estimated. $D_{max}$, particle cross-sectional area $A$, and the perimeter area $P$ are derived in the unit of pixel number. For the calculation of $A$ and $P$, only particles that do not touch the edges of the OAP are used (Crosier et al., 2011). From $A$ and $P$, we calculate the complexity parameter $\chi$, which we define as:

$$\chi = \frac{P}{2\sqrt{\pi A}},$$ (4)

similar to Gergely et al. (2017) so that $\chi$ of a sphere is 1. $\chi$ was originally proposed by Garrett and Yuter (2014), who included the inter pixel variability (the variability in brightness of one pixel compared to its neighbors) in their definition, which is not available for PIP measurements. Garrett and Yuter (2014) quantify riming based on $\chi$, where rimed particles (graupel) are defined as $\chi \leq 1.35$, moderately rimed particles as $1.35 < \chi \leq 1.75$ and aggregates with negligible riming as $\chi > 1.75$.

A disadvantage of using $\chi$ to quantify riming is that it is a purely optical measure and no physical quantity. Also, it should

be taken into account that $\chi$ not only depends on a particle's shape (closely linked to its riming degree), but also its size in pixel. Depending on the resolution of the imager as well as the exact definition of a perimeter pixel (continuous line vs. only touching outside), $\chi$ values of a circle with a diameter larger than 10 pixel can range from slightly below 0.9 to 1.3. Particle features finer than the resolution of the imager are not captured. This leads to smaller ratios of perimeter to area than for the same particle observed with a higher resolution imager. For better visualization, the reader may imagine a fractal shaped

snowflake: the higher the resolution of the snowflake image, the larger the perimeter (resulting in an infinitely large perimeter for an infinitely high resolution). In turn, larger particles have larger $\chi$ than smaller particles of the exact same shape captured by the same imager. Therefore, we take $D_{max}$, $A$, and $P$ in the unit of pixel number to account for the different resolutions of CIP and PIP.

We use the same data set of simulated rimed aggregates from Maherndl et al. (2023a) to relate particle complexity $\chi$ and

size to $M$. Only taking simulated aggregates of dendrites, we calculate $\chi$ from the average perimeter and area pixel counts over projections in the xy, yz and zx plane, where one pixel corresponds to a square with 20 μm side lengths. We then derive an empirical relation with $R^2 = 0.94$ of $\chi$ depending on $M$ and $D_{max}$ in pixel resulting in:

$$\chi = 1.33 - 0.00243 \cdot \log_{10}(M) \cdot D_{max} + 0.000171 \cdot D_{max} - 0.0854 \cdot \log_{10}(M).$$ (5)

$\chi$ is calculated from CIP and PIP measured $P$ and $A$ for each detected particle. $M$ is then calculated from $D_{max}$ and $\chi$ for

each particle:

$$\log_{10}(M) = \frac{1.33 - \chi + 0.000171 \cdot D_{max}}{0.00243 \cdot D_{max} + 0.0854}.$$ (6)





The detection efficiency of particles that do not touch the edges of the OAPs is size-depended: larger particles are more likely to touch the edge and therefore less likely to be detected than smaller particles. To account for this, we derive weighting factors for CIP and PIP, respectively, by comparing the count of total particles detected (including particles that touch edges) to the count of particles that do not touch edges. The weighing factors are derived for particle size bins from 10 to 65 pixel in 5 pixel bins (see Tab. B1 in Appendix B). From the calculated $M$, we obtain the weighted average for 1 s time steps. Then, a rolling average of 30 s is applied to make the results comparable with the $M$ retrieval described in the previous section. We only consider particles with diameters larger 14 pixel, which correspond to 210 µm for the CIP and 1400 µm for the PIP. The threshold of 14 pixel was chosen such that 99 % of Polar 6 CIP and PIP measured particles with $\chi$ smaller than 1 lie below the threshold and are therefore sorted out for the analysis. $\chi$ values smaller than 1 suggest shapes rounder than a sphere and are due to the low pixel resolution. This leaves us with a gap in the size range from about 1.0 to 1.4 mm. Evidently, only a subset of particles detected by CIP and PIP can be used to calculate $M$ raising the questions how many particles per second are enough to achieve reasonable results. By comparison to the combined method in addition to manual inspection of CIP and PIP images, we find that in sum at least 7 particles per second need to be observed for reliably calculating $M$ and thus we discard data with lower counts.

We classify particles with $M \geq 1.0$ as heavily rimed (graupel; Fig. 3.2 (b)). $M$ values larger 1.0 are physically possible and indicate rime densities larger than assumed in the aggregation and riming model ($\rho_{\mathrm{rime}} = 700$ kgm$^{-3}$). Particles with $M < 0.01$ are classified as unrimed or having negligible riming, due to their similar behavior to unrimed particles in Maherndl et al. (2023a). In between, we call particles with $0.01 \leq M < 0.1$ lightly rimed and with $0.1 \leq M < 1.0$ moderately rimed (Fig. 3.2 (a), right). Unrimed particles (Fig. 4 (a), left) have much more complex shapes and therefore larger $\chi$, than more heavily rimed ones (Fig. 4 (a), right), which are almost spherical ($\chi$ close to 1). Fig. 4 (b) shows the size dependency of $\chi$. $\chi$ for the most heavily rimed particles, which reach $M$ of about 0.87, is close to 1.33. Not shown are $\chi$ values of in situ measured cloud particles, which span values from about 0.7 to 5.0 with the majority of data (95 %) in the range of 1.0 to 3.0 for the CIP and 0.8 to 2.0 for the PIP.

## 4 Results and discussion

To investigate the performance of both methods, we first compare $M$ results for collocated flight segments and relate $M$ to meteorological as well as cloud micro- and macrophysical parameters (Sect. 4.1). We then repeat the analysis for in situ method results derived for the complete Polar 6 data set (Sect. 4.2) and present two case studies (Sect. 4.3, 4.4).

### 4.1 Comparison of both methods during collocated flight segments

Assuming particles with $M < 0.01$ having negligible riming, we derive average rimed fractions of 77 % and 75 % over all collocated flight segments with the combined and the in situ method, respectively. These numbers appear quite high, however, they depend heavily on the rimed vs. unrimed threshold that is chosen; if we assume $M < 0.05$ to be unrimed, we get 11 % and 9 %, respectively. 74 % and 73 % fall in range $0.01 \leq M < 0.1$ for the combined and in situ method, respectively, and only

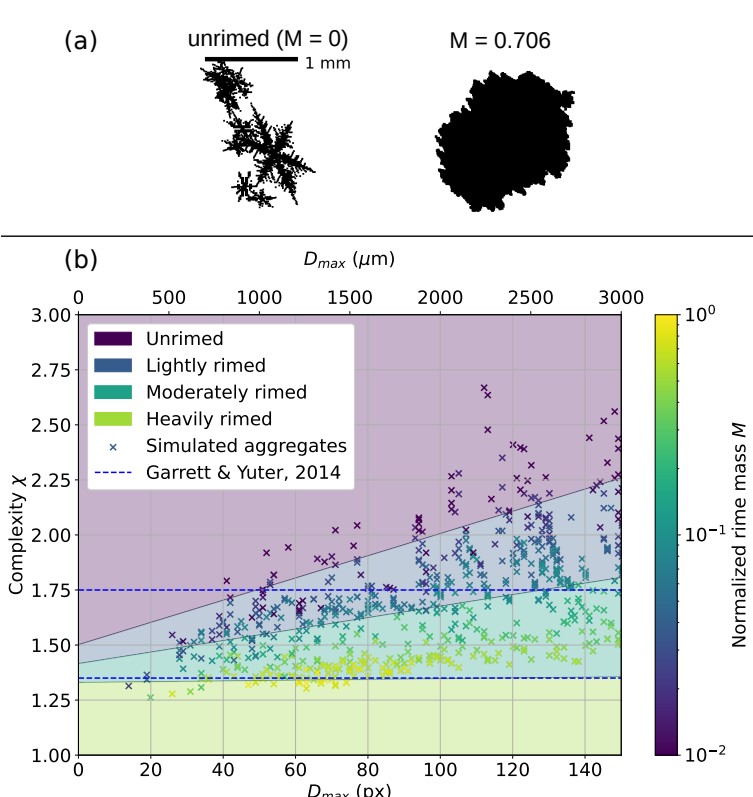

**Figure 4.** (a) Example simulated particles: unrimed dendrite aggregate (left) and moderately rimed dendrite aggregate (right). (b) Complexity $\chi$ of simulated dendrite aggregates with different amounts of riming versus their size $D_{\mathrm{max}}$ in pixel. 1 pixel corresponds to the resolution of the cubic elements (20 µm) that the simulated ice particles are composed of. Their normalized rime mass $M$ is color coded. $\chi$ thresholds for graupel (1.35) and rimed particles (1.75) from Garrett and Yuter (2014) are included as blue dashed lines. Grey lines separating differently colored areas indicate isolines of $M$ calculated with Eq. (6): $M = 0.01$ between unrimed and lightly rimed, $M = 0.1$ between lightly and moderately rimed and $M = 1.0$ between moderately and heavily rimed (graupel).





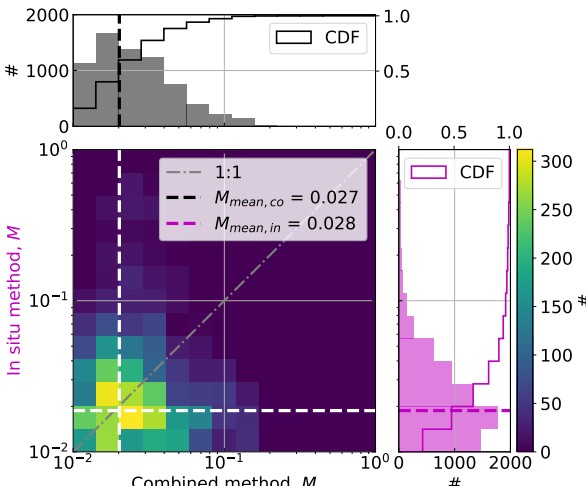

**Figure 5.** 2D histogram of $M$ derived with combined (x-axis, black) and in situ (y-axis, magenta) methods in logarithmic units during collocated flight segments. Individual histograms and cumulative distribution functions (CDF) are included in black for the combined (top panel) and in magenta for the in situ method (right panel). Respective medians are plotted as dashed lines.

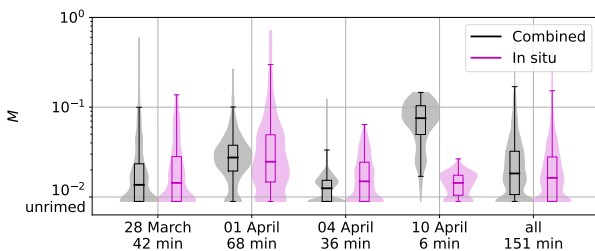

**Figure 6.** Box plots superimposed violin plots showing distributions of $M$ in logarithmic units derived with combined (black) and in situ method (magenta) for collocated flight segments on the respective flight day and in total for all regarded collocations. Approximate collocated flight time in minutes is included.

3 % and 5 % have $M \geq 0.1$. The mean of $M$ is 0.028 and 0.027, the median is 0.018 and 0.016 and the 25 % to 75 % quantile

305   ranges are 0.011 to 0.032 and 0.009 to 0.028 for combined and in situ method, respectively.

     Figure 5 shows a 2D histogram of combined and in situ method results of $M$ as well as their respective $M$ distributions. A high density of data points lies close to the 1:1 line, but data point per data point a perfect agreement cannot be expected: although we match remote sensing and in situ data points as best as possible, there still remain offsets in time (less than 5 min) and space (less than 5 km). Additionally, radar and in situ probes have different measurements volumes. Still, the similarity in

310   addition to the close agreement of means, medians, and quantile ranges gives us confidence that we achieve agreement with both methods and both can be used to quantify riming. Mean error (ME) and root mean square error (RMSE) are -0.0077 and 0.032 for the point by point comparison.





We see simlar results when comparing the individual flights except for 10 April (Fig. 6). Manual inspection of CIP and PIP images shows a high proportion of rimed particles during the collocated segment on 10 April (not shown), which is in agreement to the combined method. However, these particles appear to predominately have sizes around 1 mm – large enough to often touch edges in CIP images, but too small to be able to calculate $\chi$ from PIP images. However, collocated flight time with high enough particle counts for the in situ method results only in about 6 minutes of data therefore contributing less to the overall results than the other flights shown here.

Figure 7 gives an overview of the occurrence of riming depending on (a)-(d) Polar 6 nose boom measurements of air temperature $T$, (e)-(h) Nevzorov probe LWC, (i)-(l) MiRAC-A retrieved LWP, and (m)-(p) the position of Polar 6 in cloud (cloud bottom = 0, cloud top = 1). CTH is determined from AMALi, while cloud bottom height is determined from radar measurements, where cloud bottom is the lowest signal above 150 m. If there is a continuous signal from 150 m to the flight altitude of Polar 6, then cloud bottom is set to 150 m. Note that the liquid cloud base which is commonly used when using ground-based remote sensing, is not available for airborne measurements. Our cloud definition includes precipitation falling out of the cloud liquid layer so that a multi-layer clouds connected by precipitation would be treated as a single cloud. During the collocated flight segments used in this study, no separate cloud layers were observed by the radar above Polar 6. The average rimed fractions derived with both methods show a similar behavior for all parameters and lie on average within 7 percentage points of each other. Linear medians match within a factor of 0.3 of each other.

When analyzing the relation of riming with temperature, moderate riming also occurs at low temperature below -15 °C. Both methods show larger disagreement and (local) minima of riming between -10 and -15 °C. This coincides with the so called dendritic growth zone, where aggregation is favored (Takahashi et al., 1991; Takahashi, 2014). Complex aggregated forms can appear round when viewed from certain angles and imaged with a limited resolution. This might lead to an overestimation of riming with the in situ method. Note that only the temperature at the point of observation is available, not where – at potentially colder temperatures – the riming process itself took place. The disagreement above -10 °C stems in part from the inclusion of the 10 April data, which makes up about 40 % of data above -10 °C. On 10 April the in situ method underestimates riming due to rimed particles with sizes in the gap range between both probes.

There is no clear dependence of riming on LWC. The rimed particles could easily have undergone riming in a SLW layer above and fallen out to a place in the cloud with little to no SLW.

For LWP, median $M$ and rimed fractions increase with increasing LWP up until 50 $\mathrm{gm}^{-2}$ and decrease in the two highest LWP bins. This decrease could be due to limited sampling as the bins contain less than 500 data points. Overall, the agreement between both methods is very good with rimed fraction agreeing on average within 3 percentage points below and 11 percentage points above 100 $\mathrm{gm}^{-2}$. The discrepancy above 100 $\mathrm{gm}^{-2}$ can in part be explained by the high proportion of 10 April data (about 40 %).

Riming fractions agree within 2.7 percentage points for in cloud position above 0.2 (meaning Polar 6 is flying higher than the lowest 20 % of the cloud). Below 0.2, rimed fractions derived by the in situ method are on average 19.5 percentage points lower than by the combined method. However, median $M$ agree within a factor 0.29 above and 0.17 below 0.2. Because our definition of a cloud includes precipitation below, low cloud positions might be below the liquid cloud base. If this is indeed





**Figure 7.** Occurrence of riming during collocated flight segments derived with combined (black) and in situ method (magenta) depending on (a)-(d) Polar 6 noseboom temperature in °C, (e)-(h) Nevzorov probe measured LWC in $\mathrm{gm}^{-3}$, (i)-(l) MiRAC-A retrieved LWP in $\mathrm{gm}^{-2}$, and (m)-(p) position of Polar 6 in cloud (0 meaning bottom of cloud, 1 meaning top of cloud). Bin sizes are 2 K, 0.02 $\mathrm{gm}^{-3}$ (0.005 $\mathrm{gm}^{-3}$ below 0.02 $\mathrm{gm}^{-3}$), 20 $\mathrm{gm}^{-2}$, and 0.05 respectively. The first column shows the amount of data per bin. The second column shows rimed fraction assuming $M < 0.01$ to be unrimed derived with combined (black squares) and in situ method (magenta circles). The third and fourth columns show 2D histograms of $M$ results for combined and in situ method, respectively, including medians for each bin in white. Medians and average rimed fractions are only shown, when there are more than 100 data points per bin. Nevzorov probe data is only available in April.





the case, we expect the falling particles to be larger and heavier than the particles in the cloud above. The detection efficiency of cloud probes is worse for particles close to the upper end of their size range, even if we count particles that touch edges (as is

done in the PSD calculation). Therefore the higher rimed fractions obtained by the combined method could be due to missing large particles in the PSD that the radar can see. The retrieval would then overcompensate by increasing $M$, resulting in a higher amount of rimed particle populations. However, using running averages of 60 instead of 30 s shifts the rimed fractions below 0.2 only slightly closer together (agreement within 18.8 percent points; not shown). Additionally, sublimation below cloud result in higher $Z_e$ uncertainties. Disagreement between both methods is higher, when Polar 6 is flying near the top of

the radar signal (Fig. C1) due to the higher variability of measurements as we show in Appendix (C).

## 4.2   In situ only flights

Here, we extend the analysis to periods only covered by the in situ aircraft to demonstrate that the collocated measurements are representative of the complete Polar 6 data. Even though a large, unique data set of collocated, airborne measurements was collected in the Arctic during HALO-(AC)[3], the total number of in situ cloud measurement time exceeds the collocated

measurement time by a factor of 5. Figure 8 analyses the dependence of temperature and LWC on $M$ as shown in Fig. 7 for the collocated data set. The position of Polar 6 in-cloud as well as MiRAC-A retrieved LWP must be omitted due to the missing Polar 5 remote sensing information. The average rimed fraction assuming $M < 0.01$ ($M < 0.05$) to be unrimed is 57 (11) % with a mean $M = 0.026$, median $M = 0.012$ and a 25 % to 75 % quantile range of 0.006 to 0.027. These values are slightly lower and indicate a slight shift towards more riming during the collocated segments.

When focusing on temperature bins with a sufficiently high number of observations, we observed decreasing riming from -8 and -16 °C with decreasing temperature (Fig. 8 (b)). The rimed fraction for all in situ flights follows a similar shape to the collocated sub-sample, albeit with a lower local maximum at -9 °C (0.79 vs. 0.98). There is a slight local minimum of median $M$ and rimed fraction at about -14 °C. Lower rimed fractions and median $M$ result for lower temperatureswhen including all Polar 6 data. Similarly, rimed fraction and median $M$ are lower for LWC below 0.05 $\mathrm{gm}^{-3}$.

Differences between the in situ method results for only collocated vs. all segments are smaller when excluding Polar 6 data below 150 m as can be seen in Fig. 8: a less pronounced, but visible, local minimum of rimed fraction and median $M$ appears at about -15 °C. Also, both rimed fraction vs. LWC curves are very close, deviating by a maximum 5.4 percentage points. The better agreement above 150 m could be simply due to the higher proportion of common data points because the collocated in situ method $M$ is a subset of the in situ method $M$ derived for all Polar 6 flight segments. Another explanation could be

the influence of cloudless ice crystal precipitation ("diamond dust"). This phenomenon describes the formation of ice crystals under clear or nearly clear skies. Diamond dust typically occurs between November and mid-May in heights below 250 m over the Arctic Ocean (Intrieri and Shupe, 2004). This could shift the curve towards less riming for cold temperatures, resulting in an (almost) disappearance of the -15 °C local minimum.

We can conclude that the collocated measurements are representative of the complete Polar 6 data set above 150 m flight

altitude.



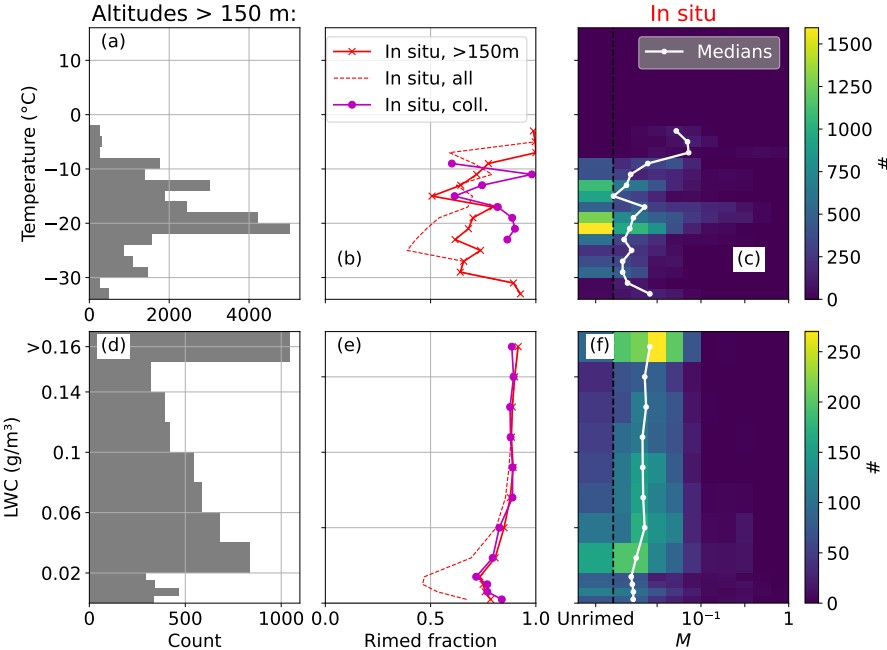

**Figure 8.** As Fig. 7 (a)-(h), but only the in situ method for all Polar 6 flights with altitudes above 150 m. Rimed fractions for all flight segments are shown as red crosses, whereas results for collocated flights are repeated as magenta circles in (b) and (e). All data including flight altitudes below 150 m are shown as red dashed lines. Nevzorov probe data is only available in April.

### 4.3 Case study 1: collocated segment, 01 April

In the following, we use a case study to demonstrate the agreement of combined and in situ method. A high pressure system north of Greenland and a strong low pressure complex north of Siberia lead to northerly and north-easterly winds almost parallel to the ice edge in the Fram Strait, where the measurements were performed. The movement of cold air from the colder

sea ice north of the Fram Strait to the warmer ocean resulted in the formation of cloud streets, which can be seen in Fig. 9 (a). Walbröl et al. (2023) could identify this cold air advection as a strong marine cold air outbreak (MCAO) that lasted from 01 to 02 April. On 01 April, Polar 5 and Polar 6 conducted collocated flights, crossing the Fram Strait perpendicular to the cloud streets from 7.6 °E to 1.5 °E back and forth three times. Clouds were thicker, more pronounced and extended higher over the open ocean on the eastern side of these segments than close to the MIZ and were absent over sea ice. Polar 5 stayed at a

constant altitude of 3 km, while Polar 6 performed predominately staircase patterns measuring in and above clouds. Here, we show a short segment where Polar 6 flew inside clouds from west to east on the eastern side of the measurement area close to 7.6 °E, while Polar 5 flew above. Then, both aircraft turned and flew the same way back westward. Excluding the turn, the horizontal distance between both airplanes ranged from 48 m to 2.7 km and was on average 1.2 km.

A detailed view of collocated in situ and radar measurements during this segment is presented in Fig. 10. The first column

shows measurements before the turn (aircraft flying from west to east, about 11:08 to 11:18 UTC), while the second column





shows measurements after the turn (east to west, about 11:25 to 11:35 UTC). We cut out the turn due to unreliable measurements and/or collocation matching when the radar is tilted due to the aircraft roll. In-cloud temperatures decreased with height ranging from $-22$ to $-15\,°C$ in the measured area (Fig. 10 (e) and (f)). The cloud's roll structure is clearly visible in the radar measurements: $Z_e$ shows periodic streaks of high and low values (Fig. 10 (c) and (d)), which can also be seen in the averaged (moving over 30 s), corrected $Z_e$ at the altitude of Polar 6 (Fig. 10 (a) and (b)). $D_{32}$ is the proxy for the mean mass-weighted diameter (e.g. Maahn et al., 2015) and is defined as the ratio of the third to the second measured PSD moments $M_3/M_2$ assuming a typical value of 2 for the exponent $b$ of the mass-size relation (e.g., Mitchell, 1996). $D_{32}$ calculated from the 30 s running average of the combined in situ PSD (Fig. 10 (g) and (h)) shows gaps when Polar 6 was flying close to cloud top (before the turn) and streaks of high $Z_e$ appear to correlate with increases in $D_{32}$. Nevzorov probe measurements (Fig. 10 (i) and (j)) show that the sampled cloud was mixed-phase with LWC being in general slightly higher close to cloud top.

We see good agreement when looking at mean, median, and quantile ranges of $M$ derived with combined and in situ method before and after the turn. The combined method results in median (mean) $M$ of 0.024 (0.031) before and 0.024 (0.027) after the turn, while the in situ method gives median (mean) $M$ of 0.025 (0.027) before and 0.017 (0.025) after the turn. 25 % to 75 % quantile ranges are 0.016 to 0.033 and 0.017 to 0.032 for the combined method before and after the turn, respectively. Quantiles range from 0.016 to 0.033 and 0.014 to 0.028 for the in situ method.

However, when comparing the time series of $M$, we see a much better agreement after than before the turn. We assume that the discrepancy before the turn is due to the Polar 6 measurements being close to the upper edge of the cloud. As discussed in Appendix C, agreement between both methods is worse close to the upper edge of the radar signal likely due to the higher variability and larger gradients of cloud properties. Even slight horizontal offsets of Polar 5 and Polar 6 in addition to the different measurement volumes of radar and cloud probes can result in disagreements between radar and in situ probes. Close to the upper edge of the cloud this can result in the radar detecting a gap in cloud while the in situ probes measure a particle concentration larger zero or vice versa. Apparently, the running averages of 30 s on both data sets cannot completely resolve this problem. In addition, median particle count drops from 22 before the turn to 17 after the turn, resulting in the in situ method being less reliable there as well.

After the turn, combined and in situ results for $M$ show better agreement as Polar 6 was flying deeper in the cloud under more homogeneous conditions. Both methods show an almost periodic in- and decrease of $M$ with (almost) matching maxima and minima in extent and location. Compared to $Z_e$ (Fig. 10 (b) and (d)), high $M$ correlate with high $Z_e$ indicating that riming plays a dominant role in MPC variability.

CIP and PIP images taken at $7\,°E$ after turn are presented in Fig. 10 (m) and (n) and show a mixture of small liquid drops, pristine plates and a high proportion of rimed (aggregated) dendritic ice particles explaining the peak in M for both methods.

### 4.4 Case study 2: in situ only segment, 08 April

We now present an in situ only case study, to show the occurrence of rimed particles in a cloud with low LWC. Several convergence lines formed in the eastern Fram Strait due to the north-easterly wind flow being disrupted by the land mass of



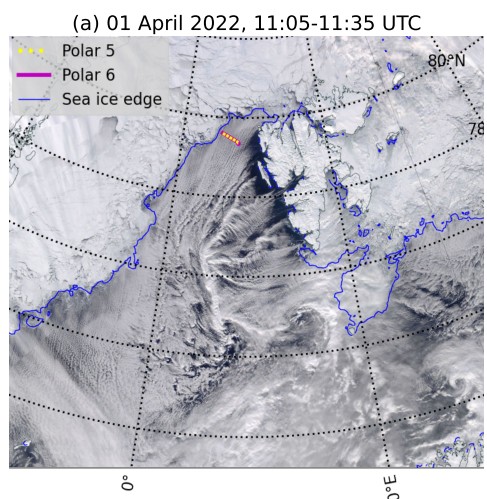
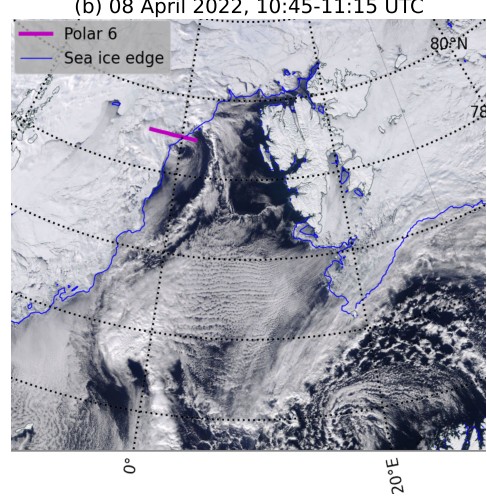

**Figure 9.** MODIS Terra reflectance images (NASA worldview) from (a) 01 April and (b) 08 April. The flight tracks of Polar 5 (yellow) and Polar 6 (magenta) as well as the sea ice edge (15 % SIC) of the same day are included.

Svalbard. A Polar Low formed (see Fig.9 (b)) and Polar 6 crossed clouds just north of the Polar Low during the presented flight

segment (Walbröl et al., 2023).

Fig.9 (b) shows the flight path of Polar 6 during the studied section, where the aircraft flew from east to west from 10:45 to 11:15 UTC, crossing into the MIZ at about 10:53 UTC. Clouds were thin, especially over the broken sea ice, where haze was detected.

Fig.11 gives a detailed overview of the flight segment and the obtained $M$ results. The aircraft started the segment in the

east at an altitude of about 1600 m where it flew until 10:50 UTC, then descended for about 5 min and flew at about 1000 m until 11:10 UTC, then ascended to 1200 m during the remainder of the segment (Fig. 11 (a)). The temperature increased from about $-16$ to $-13\,°\mathrm{C}$ during the decent. In the beginning of the segment, high concentrations of small particles, which were almost entirely liquid droplets, were detected (Fig. 11 (b) and (c)). After the descent through the liquid layer, the fraction of (large) ice particles increases resulting in an increase in $D_{32}$ and from about 10:55 UTC onward, no liquid was detected

anymore. Nonetheless, rimed particles were measured as can be seen in Fig. 11 (e) and (f) showing example CIP and PIP measurements at about 11:04:30 and 11:06.30 UTC, respectively. Fig. 11 (d) shows $M$ derived with the in situ method, where gaps indicate periods with low particle counts and therefore poor quality $M$. From 11:05 to 11:07 UTC, $M$ increases rapidly. This is confirmed by looking at CIP and PIP images: Fig. 11 (e) shows unrimed dendrites and pristine plates, while Fig. 11 (f) shows a large percentage of moderately and heavily rimed, spherical particles besides the pristine plates. These particles

must have undergone riming in the liquid layer above and fallen out due to their increased mass. The median (mean) $M$ over the segment is 0.016 (0.026), therefore on average the detected particle population is slightly rimed. We estimate the LWP by integrating LWC measurements during the descent to be about $26.6\,\mathrm{g m^{-2}}$, which is below LWP values typically assumed to be required for riming (e.g. Moisseev et al., 2017). Possible explanations of why there are moderately rimed particles even though



**Figure 10.** Collocated flight segments from 01 April 11:05-11:35 UTC before (first column) and after turn (second column). The longitude axis is reversed for the after turn segment to visualize time passing on the x-axis. (a)-(b) MiRAC measured and corrected reflectivity $Z_e$ in flight altitude of Polar 6; (c)-(d) MiRAC measured reflectivity $Z_e$, AMALi CTH, and Polar 6 flight altitude; (e)-(f) Polar 6 noseboom temperature (green) and MiRAC-A LWP (blue); (g)-(h) mass-weighted diameter $D_{32}$ derived from the 30 s running averaged combined in situ PSD; (i)-(j) Nevzorov probe LWC (blue) and TWC (black); (k)-(j) $M$ from combined (black) and in situ method (magenta) including uncertainty estimates (combined: OE standard deviation, in situ: 30 s running standard deviation); (m) example CIP; (n) PIP images.



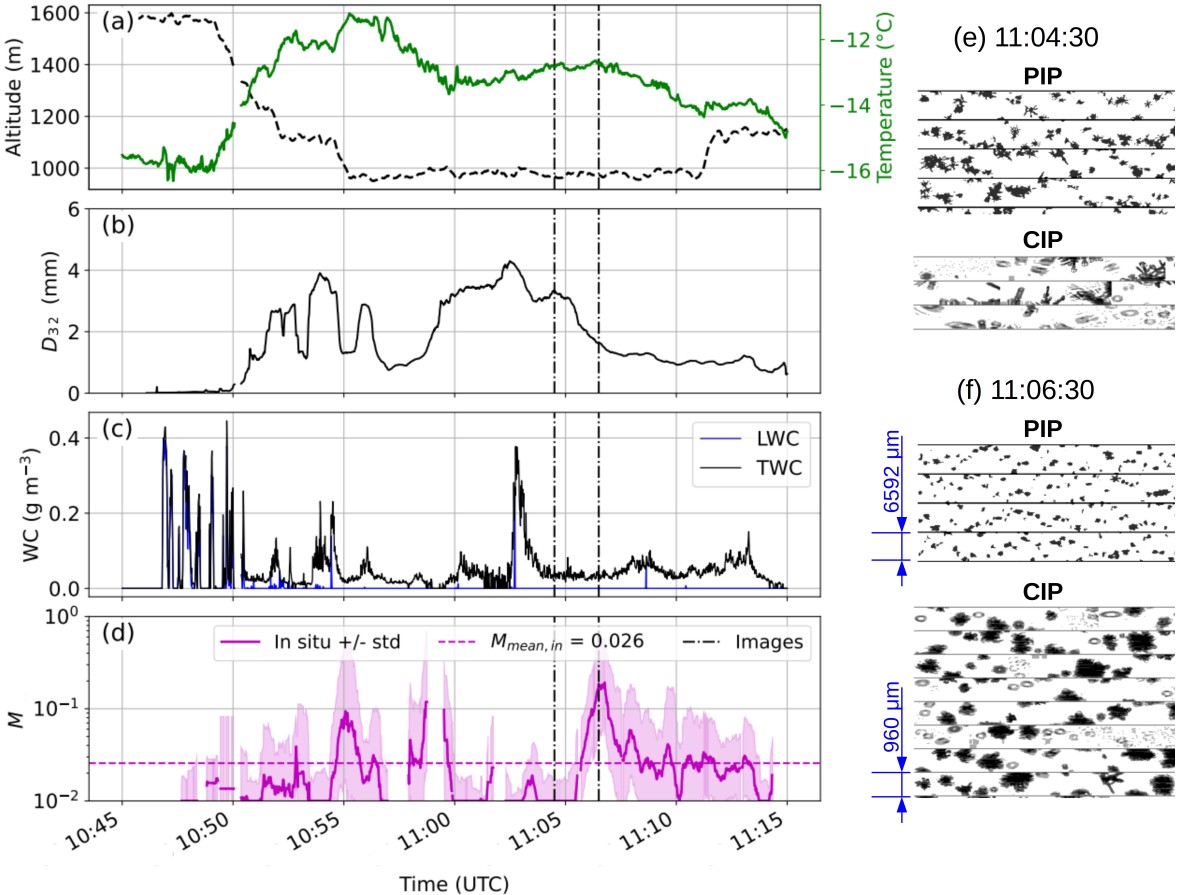

**Figure 11.** Similar to Fig. 10, but for in situ only flight segment from 08 April 10:45-11:15 UTC. Missing panels due to missing Polar 5 remote sensing observations. $M < 0.01$ is set to 0.01 to be able to show unrimed particle populations on the logarithmic scale.

LWP is low, could be seeding from a cloud layer above, turbulence or updrafts as proposed in Fitch and Garrett (2022) above
the flight altitude or simply higher LWP than estimated from the cloud profile.

## 5  Conclusions

In this study, we present two methods to quantify riming with the normalized rime mass $M$ using airborne in situ and re-
mote sensing observations. We apply both methods to data collected during the HALO-(AC)³ field campaign performed in
March/April 2022. One objective of HALO-(AC)³ was performing collocated flights with up to three aircraft. We focus on the
research aircraft Polar 5 and Polar 6, which collected closely spatially collocated and almost simultaneous in situ and remote
sensing observations west of Svalbard.



The first method takes advantage of these collocated flight segments to derive $M$. We developed an Optimal Estimation algorithm to retrieve $M$ from a combination of radar and in situ measurements by matching measured with simulated radar reflectivities $Z_e$ obtained from observed in situ particle number concentrations. As forward operator we use the Passive and Active Microwave radiative TRAnsfer tool (PAMTRA) which includes empirical relationships of $M$ and particle properties from Maherndl et al. (2023a) for estimating particle scattering properties. The latter are obtained via aggregation and riming model calculations.

With the second method, $M$ can be derived from in situ measured particle shape alone. We calculated the complexity $\chi$ of in situ measured particles, which relates particle perimeter to area. Further, we derived $M$ from empirical relationships that were again obtained from synthetic particles. However, we find that this method is only reliable when sufficient numbers of particles large enough to calculate meaningful $\chi$ are detected with the in situ probes. A threshold of 7 particles per second appears to result in a good performance.

We compare the obtained $M$ derived by both methods: combined and in situ methods result in median (mean) $M$ of 0.018 (0.027) and 0.016 (0.028) during collocated segments, $M$ distributions look remarkably similar, and the highest concentration of data lie close to the the 1:1 line when comparing collocated data point pairs directly. Looking at each flight with collocation individually, we find similar results except for 10 April when the combined method shows higher $M$ than the in situ method. By visual inspection of CIP and PIP images for the 6 minutes of collocated measurements, we find the higher $M$ predicted by the combined method to be closer to the truth. Likely, the in situ method performs worse, because a significant amount of rimed particles fall into the size range that cannot be used, i.e. particles are too large for the CIP but too small to derive $\chi$ from PIP.

In addition, we analyzed the dependence of $M$ on air temperature, LWC, LWP, and the position of Polar 6 in the cloud. Rimed fractions (assuming $M < 0.01$ to be unrimed) agree on average within 7 percentage points. With either method, we do not find a clear relation of LWC and riming during the collocated segments. LWP shows a positive correlation with riming below 130 $\mathrm{gm^{-2}}$. We confirm findings from Fitch and Garrett (2022), which show that riming also occurs in Arctic clouds with low LWP. Both methods show a decrease in riming at about -15 °C, which corresponds to the dendritic growth zone (Takahashi et al., 1991; Takahashi, 2014). Close to the upper edge of the radar signal (cloud top as seen by the radar), the methods disagree, especially when comparing data point per data point. The combined method shows higher rimed fraction and $M$ than the in situ method (Fig. C1). We think that this is likely due to the higher variability of cloud properties at cloud top resulting in less tolerance of the results to the collocation of Polar 5 and Polar 6. Disagreement is also larger close to cloud bottom, which includes precipitation below the cloud, due to detection of the liquid cloud base being unavailable from the aircraft measurements. We think that large particles that are missed by the cloud probes due to detection efficiency but seen by the radar might be the reason for higher riming fractions from the combined method.

Two case studies are presented to show 1) the performance of both methods during a collocated flight segment (case study 1) and 2) an interesting case of rimed particles observed over sea ice in a cloud segment where no liquid droplets were observed (case study 2). The first case study displays a cold air outbreak. It shows that we achieve good agreement as long as measurements are continuous, which is more often the case when Polar 6 is flying deeper in the cloud. $M$ appears to increase



and decrease periodically in correspondence with $Z_e$, indicating that riming plays an important role in the $Z_e$ variability, which is commonly observed in Arctic MPC. For the second case study, we observed a MPC close to a Polar low on 08 April with a layer of liquid droplets at cloud top and rimed particles beneath. LWC measurements are (close to) zero in a segment with
heavily rimed particles. Therefore, the particles must have been rimed in the liquid layer above and fallen out.

With both methods, we derive average $M$ over the particle population observed at a given time step. However, we often observed mixtures of pristine and rimed particles of different sizes during the campaign. While we correct the in situ method $M$ accounting for the size depended detection efficiency of CIP and PIP, we are still left with a size gap between probes. Because $Z_e$ is more sensitive to large particles, $M$ derived by the combined method is likely skewed towards the right tail of
the PSD. In future studies, the in situ method can be adapted to derive size distributions of $M$ (given the particle count per bin is sufficiently large) to compensate this.

The presented methods provide tools to better quantify riming in MPC from airborne observations. This allows to study external drivers and the variability of riming.

*Data availability.* Processed in situ and radar data as well as MiRAC-A LWP and AMALi CTH from the HALO-(AC)[3] campaign are cur-
rently being prepared for publication in PANGAEA. The data can (already) be easily accessed by the python package ac3airborne (Mech et al., 2022b). Raw in situ cloud data recorded by the CDP, CIP and PIP are archived at the German Aerospace Center and are available on request. The data set of simulated rimed aggregates generated for Maherndl et al. (2023a) is available at https://doi.org/10.5281/zenodo.7757034 (Maherndl et al., 2023b).

**Appendix A: Validation of the combined method with synthetic data**

To approximate errors of the combined method $M$ retrieval, we present results obtained for synthetic data. We use the simulated rimed dendrite aggregates from Maherndl et al. (2023a) binned into 10 logarithmic $M$ bins from $10^{-2}$ to $10^0$ ("true" $M$) and linear $D_{\max}$ bins from 0 to 10 mm with bin widths of 200 μm. We apply exponential PSDs $N(D) = N_0 \exp(-\Lambda D)$ to each $M$ bin, where $N$ is the number concentration in $\mathrm{m}^{-3}$ of particles of size $D$ in m, the intercept parameter $N_0$ (in $\mathrm{m}^{-3}$) describes the overall scaling and the slope parameter $\Lambda$ controls the shape. Similar to Maherndl et al. (2023a), we derive $N_0$ with the
empirical function from Field et al. (2006) for temperatures $T$ from -25 to -1 °C in 1 K steps. We calculate $\Lambda$ from the total number of particles $N_{tot}$ with $\Lambda = N_0/N_{tot}$ and vary $N_{tot}$ from 500 to 4500 $\mathrm{m}^{-3}$ in 500 $\mathrm{m}^{-3}$ steps. This results in a total of 2250 PSDs. We use PAMTRA to calculate $Z_e$ from the PSDs with the same set up as for the observations (see Sect. 3.1). We use the exact particle masses from the aggregation and riming model results and the SSRGA parameter calculated with snowScatt (Ori et al., 2021) that were used as a reference in Maherndl et al. (2023a). The resulting $Z_e$ are assumed to be the
"truth" and referred to as $Z_{e,true}$.

We then apply the retrieval framework of the combined method using the generated PSD in the forward operator $\mathbf{F}$ and $Z_{e,true}$ as $\mathbf{y}$. To be consistent, we assume $\mathbf{x}_a = -1$ (corresponding to $M = 0.1$) as a-priori information, $\mathbf{S}_a = 1$ as a-priori uncertainty and $\mathbf{S}_y$ corresponding to a measurement uncertainty of 1.5 dB. Mass-size and scattering are parameterized with





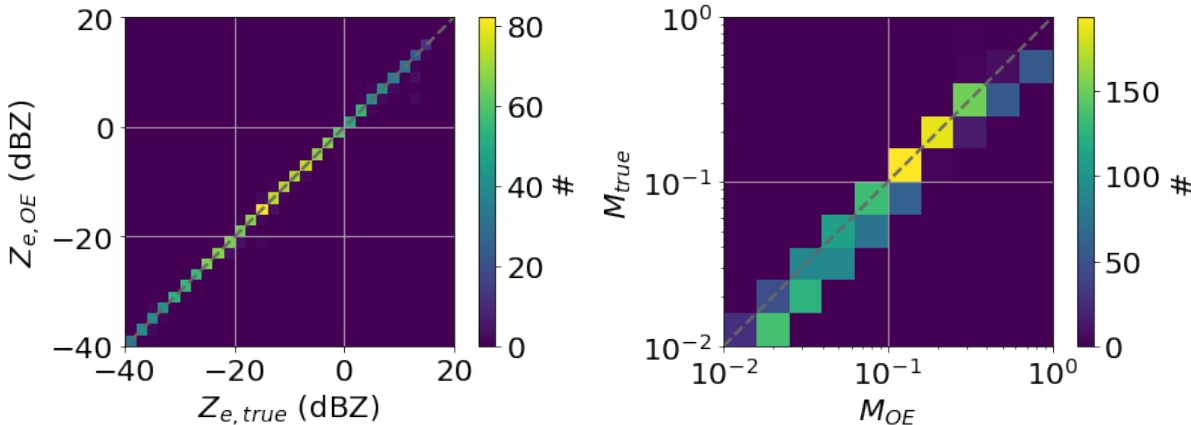

**Figure A1.** OE retrieval (combined method) results with synthetic data: (a) reflectivity $Z_{e,OE}$ vs. reflectivity $Z_{e,true}$ calculated with exact particle masses and snowScatt derived SSRGA parameter; (b) retrieved $M_{OE}$ vs. true $M_{true}$.

the riming-depended parameterization (Maherndl et al., 2023a). We therefore treat the synthetic data analogous to the in situ
observations and pretend that the mass of the particles is unknown.

Figure A1 shows (a) the resulting $Z_e$ derived with the OE framework plotted against $Z_{e,true}$ and (b) the retrieved $M$ plotted against the true $M$. OE $Z_e$ has a mean bias of -0.05 dB and an absolute mean bias of 0.09 dB compared to $Z_{e,true}$; both well within the assumed "measurement" uncertainties. $M$ is overestimated slightly for low $M_{true}$. This stems from the slight positive bias of less than 1 dB of the riming-dependent parameterization for lightly rimed particles when applying an
exponential sizes (see Fig. 10 (b) of Maherndl et al. (2023a)). In logarithmic space, the $M$ results have a mean bias of 7.7 %, which corresponds to 20 % in linear space. The uncertainty output from the OE estimation scheme results in a state space variance $\mathbf{S}_x$ corresponding to an $M$ uncertainty of 7.8 % (in the logarithmic framework).

## Appendix B: In situ method weighting factors

Tab. B1 shows the weighing factors that were derived for CIP and PIP by comparing counts of all particles to particles that do
not touch the edges of the OAPs.

## Appendix C: Limitations close to radar signal cloud top

Near the top edge of the measured radar signal, disagreement between in situ and combined method is higher (Fig. C1), which could be due to higher variability of cloud properties there: while the radar on board Polar 5 might see a gap in clouds, Polar 6 might fly a few hundred meters away in a cloudy region. Alternatively, the radar could see signatures of clouds due to its larger
footprint while the cloud probes on Polar 6 measured no particles in close proximity. Both cases do not (or rarely) occur when



**Table B1.** Weighting factors $w_{CIP}$ and $w_{PIP}$ that were derived to account for the size-dependent detection efficiency of the probes.

| Size bin (pixel) | $w_{CIP}$ | $w_{PIP}$ |
|---|---|---|
| [10, 15) | 1.53 | 1.24 |
| [15, 20) | 1.52 | 1.33 |
| [20, 25) | 1.71 | 1.42 |
| [25, 30) | 1.96 | 1.46 |
| [30, 35) | 2.35 | 1.53 |
| [35, 40) | 2.31 | 1.69 |
| [40, 45) | 2.72 | 1.62 |
| [45, 50) | 3.12 | 1.91 |
| [50, 55) | 3.64 | 2.19 |
| [55, 60) | 4.54 | 2.84 |
| [60, 65) | 6.43 | 5.35 |

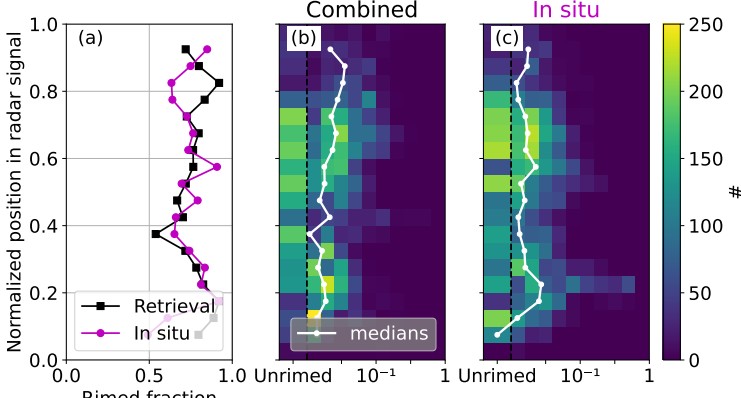

**Figure C1.** As Fig. 7 (n)-(p), but for the position of Polar 6 in the radar signal with 0 meaning cloud bottom as seen by radar (minimum 150 m due to surface clutter) and 1 meaning cloud top as seen by radar.

Polar 6 is further in the cloud where cloud properties are more homogeneous. The in situ method could also be less reliable at cloud top due to lower sample sizes. Given the available data, a concluding explanation cannot be given.

*Author contributions.* N. Maherndl developed the described methods to quantify riming, analyzed and plotted the data and wrote the manuscript. M. Maahn acquired funding and supervised the research project. M. Moser collected and processed CDP, CIP and PIP data and provided combined size distributions. J. Lucke collected and processed Nevzorov probe data. M. Mech and N. Risse collected and




processed MiRAC-A data and retrieved the LWP product. M. Mech, N. Risse and I. Schirmacher collected and processed AMALi data and retrieved the CTH product. All authors reviewed and edited the draft.

*Competing interests.* M. Maahn is a member of the editorial board of Atmospheric Measurement Techniques.

*Acknowledgements.* We gratefully acknowledge the funding by the Deutsche Forschungsgemeinschaft (DFG, German Research Foundation) for the "Arctic Amplification: Climate Relevant Atmospheric and Surface Processes, and Feedback Mechanisms" (AC)3 Project 268020496–TRR 172 within the Transregional Collaborative Research Center. Contributions by M. Moser were funded by German Research Foundation (DFG, Deutsche Forschungsgemeinschaft) under the Priority Program SPP PROM Vo1504/5-1 and by TRR - Project-ID 428312742.

Sea ice concentration data from 20 March 2022 to 10 April 2022 were obtained from https://www.meereisportal.de (grant: REKLIM-2013-04).

We thank Christof Lüpkes and Jörg Hartmann from the Alfred Wegener Institute (AWI) for providing Polar 6 noseboom air temperature measurements.



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
