# Peer review of "Quantifying riming from airborne data during HALO-(AC)3"

_EGUsphere, 2023_

## Referee Comment (RC1)

Atmospheric Measurement Techniques - Manuscript
AMT-2023-1118:
"Quantifying riming from airborne data during HALO-(AC)$^3$"
by N. Maherndl, M. Moser, J. Lucke, M. Mech, N. Risse,
I. Schirmacher and M. Maahn

**1   Summary**

This manuscript presents two new methods to quantify riming on ice particles using the normalized rime mass $M$ from airborne in situ and radar measurements. The first method combines in situ and remotely sensed radar observations in the configuration such that the radar carrying aircraft is overflying the in-situ probe carrying aircraft, while the second approach is based on in situ observations only (less demanding in terms of aircraft flight pattern). The two methods are shown to produce similar estimates of the normalized rime mass over the data collected during the HALO-(AC)$^3$ campaign that took place in the Arctic near Svalbard in Spring 2022, in a statistical sense. Two case study are further investigated. The first one (based on combined radar and in situ data) suggests that the the regions characterized by a higher normalized rime mass are related to regions exhibiting higher reflectivity values. The second one (based on in situ data only) illustrates that riming may occur in regions with low liquid water path and hence suggests that riming may occur in layers above, containing more liquid water.

**2   Recommendation**

The manuscript is clear, the methods are properly described, as well as the associated assumptions and limitations. The topic is of interest to the community and readership of AMT. There are however a few questions/issues to be clarified (see list below), and I recommend to send the manuscript back to the authors for major revisions.

**3   General comments**

1. My main concern about the evaluation of the two methods is related to Fig.5 and also Fig.7 to some extent. My take from the scatter diagram in Fig.5 and the two curves corresponding to the two methods in Fig.7 is that the two methods agree on the overall shape of the distribution of the $M$ values, in a statistical sense, but are not cofluctuating (also confirmed in case study 1, see Fig.10). In addition, most of the M values are low and the 2 methods do not seem to agree well on the very few high values, leading to a density of points that is not at all aligned with the 1-1 line in Fig.5. In Fig.7, the two methods apparently agree well for the low M value (in log scale...) but again not that much for the large M values. So I am wondering if the two methods are really in good agreement or if the data set is too unbalanced to provide a robust answer. I think this issue should be

clearly discussed in the paper, and the limitations of the comparison performed should be better emphasized.

2. In section 3.1, it is mentioned (l.217-218) that the prefactor and the exponent of the mass-size relationship are taken for dendrites. What is the influence of this choice on the retrieval of $M$ in clouds with other habits than dendrites? I did not find the discussion on this assumption.

3. In section 4.1, the vertical profiles of M are discussed and linked to environmental conditions (temperature, LWC, LWP...) but I am wondering what the uncertainties associated with the retrieved M values (and subsequently on the rimed fraction) are. And in particular if the shapes of the curves are statistically significant. As Sa is taken about 1, my gut feeling (and I may well be wrong) is that the uncertainty associated with small M values (the vast majority of the cases) is relatively large and may induce limited significance. Such uncertainties are displayed in Fig.10 for instance, why not in Fig.7? This would strengthen the analysis of the shapes of those curves (or suggest that those are not statistically significant).

4. I am mot sure I understand what it is added value of the case study 2: rimed particles are detected in regions with rather low LWC, therefore there must be layers with higher LWC above, or more generally there is not enough information about the context above the aircraft to draw any solid conclusion. So nothing original here, and I do not think it is worth being mentioned in the conclusions (see l.493-495). If this is the case, I suggest to remove the 2nd case study.

**4 Specific comments**

1. P.5, l.108: what is the influence of the choice of those parameters for the time and space consistency on the optima estimation parameters (e.g. covariances)?

2. P.7, l.171-175: I did not understand how the LWC values were estimated along the radar beam (in order to quantify the attenuation), this should be better explained.

3. P.7, l.186: I think it should be "for" instead of "to" before "our results".

4. P.9, l.235: is $Z_e$ expressed in dBz or $mm^6 m^{-3}$?

5. P.9, l.237: "to make $S_a$ more Gaussian": maybe showing a distribution (in appendix?) would strengthen the claim?

6. P.9, l.239: given that $S_y$ and $S_a$ are of the same magnitude, does the insensitivity to $S_a$ imply that the 1st term in Eq(3) is dominant and hence that F() is strongly conditioning the retrieved values?

7. P.11 l.286: it should be Fig4.b, no?

8. P.13, l.307: it seems that "a" in between "point" and "perfect" should be removed.

9. P.13, l.312: the RMSE value seems much larger than the mean value, which suggests strong uncertainty no?

10. P.14, l.14: "simIlar"

11. P. 16, l.360: ' 'Figure 8 analyses the dependence of temperature and LWC on $M$": should it be the other way around?

12. P.18, l.411: I suggest to add "in terms of temporal cofluctuations" after "agreement" to clearly emphasize on what this agreement is.

13. P.20, Fig.10: the dashed line in plots (b), (d)... is not explained in the caption.

14. P.23, l.498: "depended": should it be "depending on the"?

15. P.23, l.513: the units of $N$ and $N_0$ should be $\text{mm}^{-3}\text{mm}^{-1}$, as $N(D)dD$ is the concentration of drops of size between $D$ and $D + dD$. This is consistent with the definition of $\Lambda$ 3 lines after (in its current version, $\Lambda$ would be dimensionless).

---

## Referee Comment (RC2)

Egusphere-2023-1118 RC manuscript review:

The paper presents results from two methods for quantifying riming of ice particles in mixed-phase clouds using normalized rime mass, based on measurements collected during the HALO-(AC)[3] aircraft campaign. The performance of the two methods is compared: one method combines radar and in situ data, while the other uses only in situ data. The good agreement between these two methods allows for confident utilization of in situ data alone, especially when coincident and co-located radar data is unavailable for quantifying riming. Additionally, the authors discuss correlations between normalized rime mass, radar reflectivity, and the level of cloud liquid water content in two study cases. Considering the significance of the presented results, I believe the manuscript represents a valuable contribution to AMT. However, I have a few major comments that I suggest to be addressed before the paper is published.

**General comments**:

The manuscript requires improvement in terms of presentation and readability. Several sentences are unclear and confusing. The in situ method utilizes in situ data with a particle diameter gap ranging from 1mm to 1.4mm. However, it's not explicitly stated whether the same subset of data was employed in the combined method. If not, what would be the potential impact?

In Section 2.3, second paragraph, the text mentions the use of Tb to estimate LWP. On line 155, it states that Tb is measured from an 89 GHz passive channel. However, on line 157, the authors mention, 'Thereby, the retrieval for the LWP is based on Tb derived from simulations with the Passive and Active Microwave radiative TRAnsfer tool (PAMTRA, Mech et al., 2020), using profiles of nearby dropsondes and artificial LWPs as input.' I'm not sure which Tb value was used for the LWP estimation,

In section 3.2, only dendrite aggregates are used to estimate the relation of M and the complexity parameter and Dmax. Would it be sufficient enough to represent for other types of rimed particles? Would the relation change if other simulated aggregates are used? Also, the relation is estimated with pixel size of 20um, it brings the question of whether the outcomes would differ if the pixel size were set to match that of the Cloud Imaging Probe (CIP) or the Precipitation Imaging Probe (PIP).?

The discussion of the unit for Dmax, A, and P appears confusing. Examining Eq (4), if the size measurements are divided by the instrument's (either CIP or PIP) pixel size, one will get the pixel number. Is it correct? Regarding page 10, line 260, the statement "… the higher the resolution of the snowflake image, the larger the perimeter (resulting in an infinitely large perimeter for an infinitely high resolution)" is unclear. Does the author mean an 'infinitely large perimeter' in terms of pixel numbers? Also, in Eq(2), what is the unit for Dmax? Is it the same as in Eq (4) (i.e. pixel numbers)?

In section 4.1, I would present the second paragraph comparing M from the two method first before discussing about riming fraction with different thresholds of M. Furthermore, in Fig. 5, the authors may consider combining the two histogram plots into one (supercomposing) for a more effective visual comparison.

I find it challenging to follow Section 4. The current organization presents Section 4.1 as an analysis of data from collocated flights, Section 4.2 as all in situ only flights, Section 4.3 as an example of a case study for collocated flights, and Section 4.4 as an example of a case study for in situ only flights. The section structure could benefit from better organization. It might be helpful for the authors to consider adding explanatory text to clarify why this particular structure is necessary or beneficial.

In the case study 1 (collocated flight segment), it is not clear to me when the measurements are taken near cloud top, both methods are unreliable or only in situ method? Would the authors suggest which method should be used in those scenarios? In Fig. 10k, I don't see the correlation between time series of M from the combined and in situ methods.

**Specific comments**:

- Page 1, line 11: "… we obtain average rimed fractions of 77 % and 75 %." of what clouds or which study case?
- Page 1, line 12: "…the radar volume (about 45 m footprint diameter) …" At what distance the radar footprint (beamwidth?) is calculated? And what is the vertical resolution? (to give an ideal of a radar volume)
- Page 6, line 138: "… to be ice crystals (liquid droplets) …"  I'm not sure I understand this statement.
- Page 7, line 171: "In both cases, LWC measurements were averaged to be on a regular vertical grid with a resolution of 10 m." Could the author provide more details on the setup of the measurements, specifically how they were configured to obtain vertical LWC profiles?
- Page 7, line 173: "integrated LWC": do the author mean LWC calculated from PSD? Could the authors provide insights or comments on the accuracy comparison between Liquid Water Content (LWC) derived from Particle Size Distribution (PSD) measurements and LWC obtained through Nevzorov probe readings (LWC(Nev))?
- Page 8, line 205: "… the diameter of the smallest encompassing circle…" of what?
- Page 8, line 210: the complexity parameter is not defined until section 3.2 in page 10.
- Page 8. Line 223: "… and interpolate a_m and b_m to obtain parameters for a continuous M." I find it unclear which variable the interpolation is performed with respect to. Are a_m and b_m computed for each radar volume or for each flight segment?
- Page 9, line 228: what is the definition of the state vector x?
- Page 9, line 240: "… and Sy is the corresponding measurement uncertainty of 1.5 dB." Because Sy is a matrix, please rephrase this sentence.
- Page 9, line 240: "…in.." should be "at".
- Page 10: sentences 269-270 should be placed after Eq(6).
- Page 11,line 277: what is the spatial resolution corresponding to an averaging window of 30s?
- Page 11, line 280: "… suggest shapes rounder than a sphere" I'm not sure I understand this.
- Page 11, line 283: "By comparison to the combined method in addition to manual inspection of CIP and PIP images, we find that in sum at least …" What are the criteria the authors used in this comparison?
- Page 11, line 286: there's no Fig 3.2.
- Page 11, line 290: "Unrimed particles (Fig. 4 (a), left) have much more complex shapes and therefore larger χ, than more heavily rimed ones (Fig. 4 (a), right), which are almost spherical (χ close to 1)." This statement is true for the example shown in Fig 4 but might not be true in general, please consider to rewrite it to avoid confusion.

- Page 13, line 306: are all collocated flights segments are used to generated the histogram plots and the joint distribution in Fig. 5?
- Page 14, line 313: what is the difference in flight 10 April?
- Page 14, line 323: "… where cloud bottom is the lowest signal above 150 m". Please consider to rewrite this stanement. Cloud base could be defined based on a threshold with respect to the radar noise floor. I also wondering if 150m is good enough to avoid the ground clutter sidelobes.
- Page 14, line 329: "Both methods show larger disagreement and (local) minima of riming between -10 and -15 °C." Do the authors mean local minima of both methods and the disagreement between the two methods? If so, please consider to make this sentence clear.
- Page 14, line 334-336: If the data from April 10th is deemed unreliable for the in situ method, would the authors contemplate excluding it from the analysis. The inclusion of potentially erroneous data could compromise the validity of the analysis, especially when dealing with temperatures above -10°C.
- Page 14, line 337: "There is no clear dependence of riming on LWC." How's about the dependence of riming on TWC? I would expect to see high correlation between M and TWC, that also confirm the reliability of the methods.
- Page 15, Fig 7: legend "retrieval" should be "combined". It is hard to read the value of M. Is the black dashed line M=0.01? I couldn't find the definition of the 'normalized position in cloud' parameter used in Fig. 7m.
- Page 16, line 351: which retrieval?
- Page 16, line 352-355: "However ,…in Appendix (C)." I'm not sure what are the key points in this discussion.
- Page 16, line 360: "Figure 8 analyses the …" please rewrite.
- Page 16, line 379-380: " We can conclude that the collocated measurements are representative of the complete Polar 6 data set above 150 m flight altitude." I'm not sure I understand this statement.
- Page 18, line 412: "… close to the upper edge of the radar signal…" what is the upper edge of the radar signal?
- Page 18, line 423: what do the authors mean by "MPC variability". Variability in what parameters of MPC?
- Page 18, line 427: "… low LWC". Does the author mean "low LWC at flight altitude" ? In Fig. 11c I only see some spikes in the LWC curve or is it overlapped by the TWC curve?
- Fig 10 and 11: In addition to the D32 time series plot, it would be helpful if the authors could include a Particle Size Distribution (PSD) spectrum plot and/or a total number concentration plot. This additional information would assist in identifying segments with low particle counts. Also, what does the dash-dot line labeled as "images" mean?
- Page 22, summary section, line 485-487: In the discussion concerning the performance of the combined method near cloud base, there seems to be an implication that the method might not be reliable in this context. I would suggest a case study to explore this scenario further. Could the authors provide details on the specific limitations in cloud probe detection efficiency that might have affected the method's performance? For example, is it related to particle sizes exceeding the PIP measuring range or challenges in accurately capturing high LWC with the Nevzorov probe, or contamination of ground clutter on radar signal at low altitudes.

---

## Author Comment (AC1)

**Quantifying riming from airborne data during HALO-(AC)³**
**Response to the reviewers**

Nina Maherndl, Manuel Moser, Johannes Lucke, Mario Mech, Nils Risse, Imke Schirmacher, and Maximilian Maahn

December 8, 2023

*Original Referee comments are in italic*

> manuscript text is indented, with added text underlined and removed text crossed out.

We would like to thank the reviewers for their helpful comments. We revised the manuscript and responded to all of the reviewers' comments.

In addition, we made slight changes to improve both methods, which resulted in a shift to slightly higher $M$ values for both methods without changing the overall shape (location of maxima and minima) of $M$ results. Here, we briefly describe the changes implemented. During a calibration of the MiRAC radar against ground based radars at the AWIPEV station in Ny-Alesund, it was found that MiRAC underestimated $Z_e$ by 2 dB. Therefore the Optimal Estimation retrieval of the combined method was rerun with the corrected $Z_e$. We did not note this change in the manuscript, because the corrected $Z_e$ will be published on PANGAEA instead of the original data. Further, we improved the in situ method by implementing thresholds for equations Eq. (6) to better account for unrealistically high or low $\log_{10}(M)$ results. Very low or very high $\log_{10}(M)$ can occur due to e.g. measured $\chi$ values outside of the range covered by the simulated rimed aggregates, we used to derive the relations. We added:

> $\chi$ is calculated from CIP and PIP measured $P$ and $A$ for each detected particle. $M$ is then calculated from $D_{\max}$ and $\chi$ for each particle To avoid unrealistic values, we set all $\log_{10}(M) > 0$ to 0 and all $\log_{10}(M) < -3.5$ to -3.5. The latter threshold is chosen based on the minimum $M$ of the combined method results.

**1 Reviewer I**

*The paper presents results from two methods for quantifying riming of ice particles in mixed-phase clouds using normalized rime mass, based on measurements collected during the HALO- (AC)3 aircraft campaign. The performance of the two methods is compared: one method combines radar and in situ data, while the other uses only in situ data. The good agreement between these two methods allows for confident utilization of in situ data alone, especially when coincident and co-located radar data is unavailable for quantifying riming. Additionally, the authors discuss correlations between normalized rime mass, radar reflectivity, and the level of cloud liquid water content in two study cases. Considering the significance of the presented results, I believe the manuscript represents a valuable contribution to AMT. However, I have a few major comments that I suggest to be addressed before the paper is published.*

We thank the reviewer for the positive review and the constructive comments.

**1.1 General comments**

*The manuscript requires improvement in terms of presentation and readability. Several sentences are unclear and confusing. The in situ method utilizes in situ data with a particle diameter gap ranging from 1mm to 1.4mm. However, it's not explicitly stated whether the same subset of data was employed in the combined method. If not, what would be the potential impact?*

We thank the reviewer for the honest feedback regarding readability and sentence structure. We hope, we wear able to rewrite confusing sentences and improve readability by restructuring Section 4 and removing case study 2 (see later responses for more details). Regarding the diameter gap from 1 to 1.4 mm. This only effects the in situ, not the combined method. For the latter, we are only interested in particle counts (not shape) and can therefore use the complete data set. In fact, we have to use the complete data set, because the method is based on a closure of radar reflectivity and in situ PSD. To clarify, we added:

> This leaves us with a gap in the size range from about 1.0 to 1.4 mm. Evidently, only a subset of particles detected by CIP and PIP can be used to calculate $M$  . Therefore, the in situ method can only be applied to a subset of the in situ data that is used for the combined method. This raises the questions of how many particles per second are enough to achieve reasonable results assuming that high enough particle counts minimize effects of the data gap.

We also discuss the issue in the conclusions:

While we correct the in situ method $M$ accounting for the size  dependent detection efficiency of CIP and PIP, we are still left with a size gap between probes. $M$ results obtained with the in situ method are therefore biased towards smaller particles than 1 mm as well as larger particles than 1.4 mm. Because $Z_e$ is more sensitive to large particles, $M$ derived by the combined method is likely skewed towards the right tail of the PSD.

*In Section 2.3, second paragraph, the text mentions the use of Tb to estimate LWP. On line 155, it states that Tb is measured from an 89 GHz passive channel. However, on line 157, the authors mention, 'Thereby, the retrieval for the LWP is based on Tb derived from simulations with the Passive and Active Microwave radiative TRAnsfer tool (PAMTRA, Mech et al., 2020), using profiles of nearby dropsondes and artificial LWPs as input.' I'm not sure which Tb value was used for the LWP estimation,*

We apologize for the confusing sentence(s). We use measured Tb values from a 89 GHz passive channel. We then a apply the retrieval described in Ruiz-Donoso et al. (2020) to the measured Tb to retrieve LWP. This retrieval is based on Tb derived from simulations. We rephrased:

> From this observations, the liquid water path (LWP) is estimated over open ocean only with a temporal resolution of 1 s as described in Ruiz-Donoso et al. (2020).  The retrieval takes profiles of nearby dropsondes to calculate $T_B$  as a function of LWP measurements from simulations with the Passive and Active Microwave radiative TRAnsfer tool (PAMTRA, Mech et al., 2020). These functions are then applied to the 89 GHz $T_B$ measurements to derive LWP.

*In section 3.2, only dendrite aggregates are used to estimate the relation of M and the complexity parameter and Dmax. Would it be sufficient enough to represent for other types of rimed particles? Would the relation change if other simulated aggregates are used? Also, the relation is estimated with pixel size of 20um, it brings the question of whether the outcomes would differ if the pixel size were set to match that of the Cloud Imaging Probe (CIP) or the Precipitation Imaging Probe (PIP).?*

We added the Appendix section A ("Assumption on particle shape") to discuss the dendrite assumption for both methods. We decided to compare to assuming columns or plates, due to the temperature range of the majority of HALO-(AC)³ measurements (as is further discussed in this appendix section). We found that for the in situ method, dendrites are sufficient, both column and plate results lie within the uncertainty estimates for dendrite results.

In the development of the in situ method, we analyzed the impact of applying the relation derived for 20um resolution to coarser resolution data. We did so by decreasing

the resolution of the synthetic particles to 60 um (by grouping together three by three pixel) and found that the outcome did not differ significantly. We added:

> By applying the relation derived for synthetic particles with a 20 µm resolution to CIP and PIP measurements with 15 µm and 103 µm resolution, respectively, we assume the ice particle shape to be fractal - i.e., $\chi$ only depends on $D_{\mathrm{max}}$ in pixel (and $M$) and not $D_{\mathrm{max}}$ in a physical length unit. To check this assumption, we decreased the "resolution" of the synthetic ice particles to 60 µm by grouping together three by three pixels and applied Eq. 6. The resulting $M$ bias is 27 % and in the same range of using the original 20 µm particles (21 %).

*The discussion of the unit for Dmax, A, and P appears confusing. Examining Eq (4), if the size measurements are divided by the instrument's (either CIP or PIP) pixel size, one will get the pixel number. Is it correct? Regarding page 10, line 260, the statement "... the higher the resolution of the snowflake image, the larger the perimeter (resulting in an infinitely large perimeter for an infinitely high resolution)" is unclear. Does the author mean an 'infinitely large perimeter' in terms of pixel numbers? Also, in Eq(2), what is the unit for Dmax? Is it the same as in Eq (4) (i.e. pixel numbers)?*

Yes, we added:

> From $A$ and $P$ in the unit of pixel numberss, we calculate the complexity parameter $\chi$, which we define as:

Again, we apologize for the confusing sentence. We changed to:

> For better visualization, the reader may imagine a fractal shaped snowflake: the higher the resolution of the snowflake image, the larger the perimeter (not only in pixel numbers, but also when converting to a physical length. For any fractal shape, the length of the shape increases, with increasing resolution resulting in an infinitely large perimeter for an infinitely high resolution).

In Eq. (2), Dmax is in m. We added:

> The maximum dimension $D_{\mathrm{max}}$ is defined as the diameter of the smallest encompassing circle circle encompassing the cloud particle in m and is used to parameterize particle sizes during the whole study (only for the in situ method, we convert $D_{\mathrm{max}}$ from physical units to pixel number).

*In section 4.1, I would present the second paragraph comparing M from the two method first before discussing about riming fraction with different thresholds of M. Furthermore, in Fig. 5, the authors may consider combining the two histogram plots into one (super-composing) for a more effective visual comparison.*

We changed the structure of section 4.1 accordingly and added supercomposites of the histograms into Fig. 5.

*I find it challenging to follow Section 4. The current organization presents Section 4.1 as an analysis of data from collocated flights, Section 4.2 as all in situ only flights, Section 4.3 as an example of a case study for collocated flights, and Section 4.4 as an example of a case study for in situ only flights. The section structure could benefit from better organization. It might be helpful for the authors to consider adding explanatory text to clarify why this particular structure is necessary or beneficial.*

We restructured Section 4, added sentences to clarify the (new) order and hope it is better to follow now: We start by showing the statistical comparison of both methods during collocated flight segments. We then show a case study (formerly case study 1) to visualize biases of both methods and analyze under which conditions both methods agree also in terms of temporal confluctiations. We then show the comparison to meteorological and cloud parameter. Lastly, we extend the analysis to the (larger) in situ only data set. Case study 2 was removed due to the comment of Reviewer 2 that it is not necessary, which we agree with.

*In the case study 1 (collocated flight segment), it is not clear to me when the measurements are taken near cloud top, both methods are unreliable or only in situ method? Would the authors suggest which method should be used in those scenarios? In Fig. 10k, I don't see the correlation between time series of M from the combined and in situ methods.*

Unfortunately, both methods are less reliable near cloud top, as we discuss in the following. However, when averaged over long enough segments, they do provide reliable results in a statistical sense:

> However, when comparing the time series of $M$, we see a much better agreement in terms of temporal confluctuations after than before the turn. We assume that the discrepancy before the turn is due to the Polar 6 measurements being close to the upper edge of the cloud. As discussed in Appendix D, agreement between both methods is worse close to the highest radar range gates with cloud signals. This is likely due to the higher spatial variability and larger spatial gradients of cloud properties. Even slight horizontal offsets of Polar 5 and Polar 6 in addition to the different measurement volumes of radar and cloud probes can result in disagreements between radar and in situ probes. Close to the upper edge of the cloud this can result in the radar detecting a gap in cloud while the in situ probes measure a particle concentration larger zero or vice versa. Apparently, the running averages of 30 s on both data sets cannot completely resolve this problem. In addition, median particle count increases from 17 before the turn to 22 after the turn, resulting in the

in situ method being less reliable before the turn as well. Therefore, near cloud top, both methods are less reliable in a spatio-temporal sense. They do, however, both produce reliable estimations of $M$ in a statistical sense.

**1.2 Specific comments**

*- Page 1, line 11: ". . . we obtain average rimed fractions of 77 % and 75 %." of what clouds or which study case?*

We mean over all collocated segments, sorry for omitting that here. We added:

> Assuming particles with a normalized rime mass smaller 0.01 to be unrimed, we obtain average rimed fractions of 88 % and 87 % over all collocated flight segments, respectively.

(The percentage values changed due to the improvements to both methods decribed in the beginning.)

*- Page 1, line 12: ". . . the radar volume (about 45 m footprint diameter) . . . " At what distance the radar footprint (beamwidth?) is calculated? And what is the vertical resolution? (to give an ideal of a radar volume)*

To clarify, we added:

> Although in situ measurement volumes are in the range of a few cm³ and therefore much smaller than the radar volume (about 45 m footprint diameter at an altitude of 500 m above ground; with a vertical resolution of 5 m), we assume they are representative of the radar volume.

*- Page 6, line 138: ". . . to be ice crystals (liquid droplets) . . . " I'm not sure I understand this statement.*

We simply chose a cut off and assume all particles smaller the cut of to be liquid droplets and all particles larger this cut off to be ice particles, because we cannot discriminate between small droplets and small ice particles. We rephrased to:

>  We therefore assume all cloud particles with sizes larger 50 μm to be ice crystals  and all cloud particles with sizes smaller 50 μm to be liquid droplets similar to Moser et al. (2023). For the majority of low-level Arctic MPC, this is appropriate to assume (McFarquhar et al., 2007; Korolev et al., 2017).

*- Page 7, line 171: "In both cases, LWC measurements were averaged to be on a regular*

*vertical grid with a resolution of 10 m." Could the author provide more details on the setup of the measurements, specifically how they were configured to obtain vertical LWC profiles?*

Yes, we apologize for keeping this analysis step confusingly brief. We extended:

> To estimate attenuation due to liquid water, we took LWC measurements from the Nevzorov probe operated onboard Polar 6 during the temporally closest vertical cloud profile. To obtain information on the vertical structure of clouds, Polar 6 flew vertical profiles in so-called "saw-tooth patterns". These patterns were flown in addition to straight legs at constant altitudes. Saw-tooth patterns are not well suited for good quality collocated measurements with Polar 5, where straight legs are preferred. Therefore a limited number of vertical profiles are available for each flight with collocation. During each flight analyzed in this study, at least three of such "saw-tooth patterns" were collected. Whenever Nevzorov probe measurements were not available, LWC was calculated by integrating the particle size distribution (PSD) of liquid particles (¡ 50 µm) measured with the cloud probes on board Polar 6. In both cases, LWC measurements were averaged to be on a regular vertical grid with a resolution of 10 m. Here, we neglect the distance traveled by Polar 6 during the profile, assuming LWC to be constant at each height bin. This assumption likely does not hold in reality, however, no measurements with more precise information on horizontal and vertical LWC distributions are available.

*- Page 7, line 173: "integrated LWC": do the author mean LWC calculated from PSD? Could the authors provide insights or comments on the accuracy comparison between Liquid Water Content (LWC) derived from Particle Size Distribution (PSD) measurements and LWC obtained through Nevzorov probe readings (LWC(Nev))?*

Yes, we mean calculated from PSD (see changes above). Information on the accuracy of Nevzorov probe LWC and LWC calculated from PSD is included in the last paragraph of Sect. 2.2 and we chose not to repeat here:

> This assumption is based on the good agreement between Nevzorov probe LWC and LWC calculated from the PSD assuming particles smaller 50 µm to be liquid droplets where both measurements are available ($R^2 = 0.83$; Nevzorov and PSD LWC sum up to 973 and 983 g m$^{-3}$, respectively, and lie within 1 % of each other). Additionally, we do not expect this assumption will lead to significant biases due to radar reflectivities (that we simulate from in situ PSDs) being dominated by large particles.

*- Page 8, line 205: "... the diameter of the smallest encompassing circle..." of what?*

We clarified in text, that we mean the smallest circle that encompasses the ice particle:

> The maximum dimension $D_{\max}$ is defined as the diameter of the smallest  circle encompassing the cloud particle in m ...

*- Page 8, line 210: the complexity parameter is not defined until section 3.2 in page 10.*

Thanks for catching that. We decided to remove the mentioning of the complexity parameter here.

*- Page 8. Line 223: "... and interpolate a_m and b_m to obtain parameters for a continuous M." I find it unclear which variable the interpolation is performed with respect to. Are a_m and b_m computed for each radar volume or for each flight segment?*

a_m and b_m are computed for each time step, because the Optimal Estimation retrieval is applied to each time step to derive $M$. However, here we mean the interpolation that is needed to derive both a_m and b_m as a function of $M$. We rephrased:

> In addition, we consider the mass-size relation to follow a power law ($m = a_m \cdot D_{\max}^{b_m}$), and take the mass-size parameters $a_m$ and $b_m$ for dendrites from the same study. There $a_m$ and $b_m$ are given for discrete $M$, so we interpolate $a_m$ and $b_m$ to obtain parameters for a continuous $M$.

*- Page 9, line 228: what is the definition of the state vector x?*

We use the logarithm of $M$ as the state vector:

> We  choose **x** to represent $M$ in common logarithmic scale ($x = [\log_{10}(M)]$) to avoid negative values.

*- Page 9, line 240: "... and Sy is the corresponding measurement uncertainty of 1.5 dB." Because Sy is a matrix, please rephrase this sentence.*

*- Page 9, line 240: "...in.." should be "at".*

Addressing both comments, we changed to:

> **y** are the attenuation corrected $Z_e$ measurements  at Polar 6 flight altitude in dBZ and $\mathbf{S}_y$  represents the corresponding measurement uncertainty of 1.5 dB.

*- Page 10: sentences 269-270 should be placed after Eq(6).*

Done.

*- Page 11,line 277: what is the spatial resolution corresponding to an averaging window of 30s?*

We added:

> Then, a rolling average of 30 s (corresponding to 1.8-2.4 km for the typical Polar 6 flight speed of 60-80 m s$^{-1}$) is applied to make the results comparable with the $M$ retrieval described in the previous section.

*- Page 11, line 280: "... suggest shapes rounder than a sphere" I'm not sure I understand this.*

We agree this formulation does not make much sense and removed it.

*- Page 11, line 283: "By comparison to the combined method in addition to manual inspection of CIP and PIP images, we find that in sum at least ..." What are the criteria the authors used in this comparison?*

We found large disagreements between $M$ distributions for collocated segments during March 20. By inspecting in situ images manually during these segments (checking if the particles look rimed or pristine by eye), we found the combined method results to appear trustworthy. We then found that particle count was very low during March 20 and evaluated different particle count cut offs. Eventually, by comparing $M$ distributions for all collocated segments and checking a subset of in situ images during each segment, we found 7 particles per second the lowest cut-off to achieve sensible results. This lead to the complete removal of the March 20 data for the analysis.

*- Page 11, line 286: there's no Fig 3.2.*

Thanks, we changed to the correct reference.

*- Page 11, line 290: "Unrimed particles (Fig. 4 (a), left) have much more complex shapes and therefore larger $\chi$, than more heavily rimed ones (Fig. 4 (a), right), which are almost spherical ($\chi$ close to 1)." This statement is true for the example shown in Fig 4 but might not be true in general, please consider to rewrite it to avoid confusion.*

True, we rephrased:

>  In most cases, unrimed particles (Fig. 4 (a), left) have much more complex shapes and therefore larger $\chi$, than more heavily rimed ones (Fig. 4 (a), right), which are almost spherical ($\chi$ close to 1).

*- Page 13, line 306: are all collocated flights segments are used to generated the histogram plots and the joint distribution in Fig. 5?*

Yes, we added:

> Figure 5 shows a 2D histogram of combined and in situ method results of $M$ for all collocated flight segments as well as their respective $M$ distributions.

*- Page 14, line 313: what is the difference in flight 10 April?*

A large percentage of rimed particles falls into and close to the size gap of the in situ method, as we further discuss here:

> We see  similar results when comparing the individual flights except for 10 April (Fig. 6). Manual inspection of CIP and PIP images shows a high proportion of rimed particles during the collocated segment on 10 April (not shown), which is in agreement to the combined method.  These particles appear to predominately have sizes around 1 mm – large enough to often touch edges in CIP images, but too small to be able to calculate $\chi$ from PIP images.  In all further analysis steps, we exclude the April 10 data, which corresponds to 6  minutes of collocated data.

*- Page 14, line 323: "... where cloud bottom is the lowest signal above 150 m". Please consider to rewrite this statement. Cloud base could be defined based on a threshold with respect to the radar noise floor. I also wondering if 150m is good enough to avoid the ground clutter sidelobes.*

The 150 m threshold for ground clutter contamination is discussed in Mech et al. (2019), where the radar processing steps are also presented in detail. During most collocated flight segments, the radar observed a continuous signal towards the cloud and we were not able to discriminate what is still in cloud and what is precipitation below cloud from the radar observations. We agree that the statement is problematic and rephrased to:

> CTH is determined from AMALi, while  CBH is determined from radar measurements, where cloud bottom is the lowest  $Z_e$ measurement not affected by ground clutter.

*- Page 14, line 329: "Both methods show larger disagreement and (local) minima of riming between -10 and -15 °C." Do the authors mean local minima of both methods and the disagreement between the two methods? If so, please consider to make this sentence clear.*

Yes, we rephrased to:

>  Between -10 and -15 °C, (local) minima of  rimed fractions and $M$ are evident with both methods.

*- Page 14, line 334-336: If the data from April 10th is deemed unreliable for the in situ method, would the authors contemplate excluding it from the analysis. The inclusion of*

*potentially erroneous data could compromise the validity of the analysis, especially when dealing with temperatures above -10°C.*

We agree and removed the April 10 data from the further analysis steps. This is noted in the text (see response to the 10 April comment above).

*- Page 14, line 337: "There is no clear dependence of riming on LWC." How's about the dependence of riming on TWC? I would expect to see high correlation between M and TWC, that also confirm the reliability of the methods.*

We added a panel with TWC to the plot (now Fig. 9 (i)-(l)) and discuss our findings in text. However, we disagree that a relation between M and TWC is necessarily expected (see text). Aggregation can also lead to higher TWC due to large particles and we observed less riming in a temperature range (around -15°C) where large, fluffy aggregates can be expected.

*- Page 15, Fig 7: legend "retrieval" should be "combined". It is hard to read the value of M. Is the black dashed line M=0.01? I couldn't find the definition of the 'normalized position in cloud' parameter used in Fig. 7m.*

Sorry, for the wrong labeling, we changed the legend. The black dashed line is M=0.01, we added that to the figure description. Also, we added the definition of the "normalized position in cloud":

> ... t) the normalized position of Polar 6 in cloud(, which we define as the fraction of Polar 6 flight altitude minus cloud bottom height (CBH) and CTH minus CBH (therefore: cloud bottom = 0, cloud top = 1).

*- Page 16, line 351: which retrieval?*

The Optimal Estimation retrieval from the combined method. We clarified:

> The Optimal Estimation retrieval would then overcompensate by increasing M, resulting in a higher amount of rimed particle populations . However, using for the combined method.

*- Page 16, line 352-355: "However ,...in Appendix (C)." I'm not sure what are the key points in this discussion.*

We added further text to clarify:

> Averaging the in situ data for longer time spans should ensure capturing more large particles. Using running averages of 60 instead of 30 s shifts the rimed fractions below 0.2 only slightly closer together (agreement within 18.8 percent points; not shown). Additionally, sublimation below cloud result in higher $Z_e$ uncertaintiesHowever, average particles sizes increase at small

normalized positions in cloud. Median values of $D_{32}$ increase by about 150 % from 1.52 mm at 0.15-0.2 to 3.71 mm at 0.05-0.1. Disagreement between both methods is higher, when Polar 6 is flying near the top of the radar signal (Fig. D1) due to the higher variability of measurements as we show in Appendix (D).

*- Page 16, line 360: "Figure 8 analyses the ..." please rewrite.*

Done.

*- Page 16, line 379-380: " We can conclude that the collocated measurements are representative of the complete Polar 6 data set above 150 m flight altitude." I'm not sure I understand this statement.*

Because the collocated segments are only a subset of all in situ data collected during HALO-(AC)³, we wanted to check how representative our findings concerning riming are for the whole campaign. We rephrased the conclusion:

We can conclude that the collocated  flight segments are in part representative of all Polar 6 flight segments where Polar 6 flew above 150 m they show similar behavior in terms of $M$ dependence on LWC. However the collocated segments are biased towards higher amounts of rimed particles at low temperatures below -17 °C.

*- Page 18, line 412: "... close to the upper edge of the radar signal..." what is the upper edge of the radar signal?*

The cloud top as seen by the radar reflectivity measurements. Because the radar is not as sensitive to small droplets as the lidar, it often sees lower cloud tops. We added:

Close to the upper edge of the radar signal (cloud top as seen by the radar reflectivity measurements), ...

*- Page 18, line 423: what do the authors mean by "MPC variability". Variability in what parameters of MPC?*

We meant in terms of radar reflectivity observations and added:

... high $M$ correlate with high $Z_e$ indicating that riming plays a dominant role in MPC variability as observed by radar.

*- Page 18, line 427: "... low LWC". Does the author mean "low LWC at flight altitude" ? In Fig. 11c I only see some spikes in the LWC curve or is it overlapped by the TWC curve?*

As already mentioned, we removed case study 2.

*- Fig 10 and 11: In addition to the D32 time series plot, it would be helpful if the authors could include a Particle Size Distribution (PSD) spectrum plot and/or a total number concentration plot. This additional information would assist in identifying segments with low particle counts. Also, what does the dash-dot line labeled as "images" mean?*

We added PSD panels in what is now Fig. 8. We also included a description of the dash-dotted line in the figure description. This is the location, where the images in panels (o) and (p) are taken.

*- Page 22, summary section, line 485-487: In the discussion concerning the performance of the combined method near cloud base, there seems to be an implication that the method might not be reliable in this context. I would suggest a case study to explore this scenario further. Could the authors provide details on the specific limitations in cloud probe detection efficiency that might have affected the method's performance? For example, is it related to particle sizes exceeding the PIP measuring range or challenges in accurately capturing high LWC with the Nevzorov probe, or contamination of ground clutter on radar signal at low altitudes.*

We expect the problem to be large particles close to cloud base or in precipitation below cloud (which our cloud criterion includes). These particles are less likely to be detected by the PIP, or missed entirely if they are larger than 6.4 mm. Typically, there are no liquid droplets close to cloud bottom and no liquid precipitation below, therefore Nevzorov probe measurements are not the issue. Also, we expect no problems due to ground clutter (we only use measurements above the ground clutter threshold). To check our assumption, that large particles are a fault, we calculated mean, median and quantile D32 values in each normalized position in cloud bin. We found that in the lowest bins, D32 increases drastically. In the manuscript we included:

> We think that large particles that are missed by the cloud probes due to detection efficiency but seen by the radar might be the reason for higher riming fractions from the combined method. Median values of $D_{32}$ over all collocated segments increase by about 150 % from 1.52 mm at a normalized position in cloud of 0.15-0.2 to 3.71 mm at 0.05-0.1.

In case, the reviewer is interested, we include the following plot here: We do not show this figure in the study, because we don't think it provides much further information that the addition to the text above.

[Figure]

Figure R.1: Mean (red crosses, dashed), median (black circles, solid) and 25-75 % quantile range (black shaded) of $D_{32}$ in m as a function of the normalized position of Polar 6 in cloud. Similar to Fig. 9 in the main text, the data was binned in normalized position bins of 0.05. Only data in bins with more than 100 data points is shown.

---

## Author Comment (AC2)

**Quantifying riming from airborne data during HALO-(AC)³**
**Response to the reviewers**

Nina Maherndl, Manuel Moser, Johannes Lucke, Mario Mech, Nils Risse, Imke Schirmacher, and Maximilian Maahn

December 8, 2023

*Original Referee comments are in italic*

> manuscript text is indented, with added text underlined and

We would like to thank the reviewers for their helpful comments. We revised the manuscript and responded to all of the reviewers' comments.

In addition, we made slight changes to improve both methods, which resulted in a shift to slightly higher $M$ values for both methods without changing the overall shape (location of maxima and minima) of $M$ results. Here, we briefly describe the changes implemented. During a calibration of the MiRAC radar against ground based radars at the AWIPEV station in Ny-Alesund, it was found that MiRAC underestimated $Z_e$ by 2 dB. Therefore the Optimal Estimation retrieval of the combined method was rerun with the corrected $Z_e$. We did not note this change in the manuscript, because the corrected $Z_e$ will be published on PANGAEA instead of the original data. Further, we improved the in situ method by implementing thresholds for equations Eq. (6) to better account for unrealistically high or low $\log_{10}(M)$ results. Very low or very high $\log_{10}(M)$ can occur due to e.g. measured $\chi$ values outside of the range covered by the simulated rimed aggregates, we used to derive the relations. We added:

> $\chi$ is calculated from CIP and PIP measured $P$ and $A$ for each detected particle. $M$ is then calculated from $D_{\max}$ and $\chi$ for each particle . To avoid unrealistic values, we set all $\log_{10}(M) > 0$ to 0 and all $\log_{10}(M) < -3.5$ to -3.5. The latter threshold is chosen based on the minimum $M$ of the combined method results.

**2 Reviewer II**

**2.1 Summary**

*This manuscript presents two new methods to quantify riming on ice particles using the normal- ized rime mass M from airborne in situ and radar measurements. The first method combines in situ and remotely sensed radar observations in the configuration such that the radar carrying aircraft is overflying the in-situ probe carrying aircraft, while the second approach is based on in situ observations only (less demanding in terms of aircraft flight pattern). The two methods are shown to produce similar estimates of the normalized rime mass over the data collected during the HALO-(AC)3 campaign that took place in the Arctic near Svalbard in Spring 2022, in a statistical sense. Two case study are further investigated. The first one (based on combined radar and in situ data) suggests that the the regions characterized by a higher normalized rime mass are related to regions exhibiting higher reflectivity values. The second one (based on in situ data only) illustrates that riming may occur in regions with low liquid water path and hence suggests that riming may occur in layers above, containing more liquid water.*

**2.2 Recommendation**

*The manuscript is clear, the methods are properly described, as well as the associated assumptions and limitations. The topic is of interest to the community and readership of AMT. There are however a few questions/issues to be clarified (see list below), and I recommend to send the manuscript back to the authors for major revisions.*

We thank the reviewer for the positive review and the constructive comments.

**2.3 General comments**

*1. My main concern about the evaluation of the two methods is related to Fig.5 and also Fig.7 to some extent. My take from the scatter diagram in Fig.5 and the two curves corresponding to the two methods in Fig.7 is that the two methods agree on the overall shape of the distribution of the M values, in a statistical sense, but are not cofluctuating (also confirmed in case study 1, see Fig.10). In addition, most of the M values are low and the 2 methods do not seem to agree well on the very few high values, leading to a density of points that is not at all aligned with the 1-1 line in Fig.5. In Fig.7, the two methods apparently agree well for the low M value (in log scale...) but again not that much for the large M values. So I am wondering if the two methods are really in good agreement or if the data set is too unbalanced to provide a robust answer. I think*

*this issue should be clearly discussed in the paper, and the limitations of the comparison performed should be better emphasized.*

Thank you for the comment. We agree that we achieve agreement in a statistical sense, but are not in terms of temporal cofluctuations for all flight segments. We wanted to show under which condition the latter can be achieved using case study 1, but this was not stated clearly enough in the first manuscript version. We restructured Sect. 4, and added more discussion about the evaluation of the two methods throughout. In addition, we added uncertainty estimates in what was formerly Fig. 7 (now Fig. 9). The uncertainties stem from the Optimal Estimation output for the combined method, and 30 s running standard deviation calculation for the in situ method.

*2. In section 3.1, it is mentioned (l.217-218) that the prefactor and the exponent of the mass-size relationship are taken for dendrites. What is the influence of this choice on the retrieval of M in clouds with other habits than dendrites? I did not find the discussion on this assumption.*

We added the Appendix section A ("Assumption on particle shape") to discuss the dendrite assumption for both methods. We decided to compare to assuming columns or plates, due to the temperature range of the majority of HALO-(AC)[3] measurements (as is further discussed in this appendix section). We found that for the combined method, results for dendrites and plates agree within the uncertainty estimates. However, in clouds with column habits, we typically overestimate $M$ by assuming dendrites. But only a small percentage of measurements (about 10% for collocated segments) were taken in temperature regimes that favor column- over plate-like shapes. In future studies, we plan to evaluate incorporating particle habit classification results into the retrieval schemes.

*3. In section 4.1, the vertical profiles of M are discussed and linked to environmental conditions (temperature, LWC, LWP...) but I am wondering what the uncertainties associated with the retrieved M values (and subsequently on the rimed fraction) are. And in particular if the shapes of the curves are statistically significant. As Sa is taken about 1, my gut feeling (and I may well be wrong) is that the uncertainty associated with small M values (the vast majority of the cases) is relatively large and may induce limited significance. Such uncertainties are displayed in Fig.10 for instance, why not in Fig.7? This would strengthen the analysis of the shapes of those curves (or suggest that those are not statistically significant).*

As discussed above, we included uncertainty estimates in the revised figure. The shape of the curves are still significant, especially the minima, we found at temperatures of about -15°C.

*4. I am mot sure I understand what it is added value of the case study 2: rimed particles are detected in regions with rather low LWC, therefore there must be layers with higher LWC above, or more generally there is not enough information about the context above the aircraft to draw any solid conclusion. So nothing original here, and I do not think it*

*is worth being mentioned in the conclusions (see l.493-495). If this is the case, I suggest to remove the 2nd case study.*

We agree, case study 2 was not necessary and we therefore removed it. We thank the reviewer for this suggestion as we feel it really helped us to restructure Sect. 4 and improve readability of the manuscript.

**2.4 Specific comments**

*1. P.5, l.108: what is the influence of the choice of those parameters for the time and space consistency on the optima estimation parameters (e.g. covariances)?*

We do not include the uncertainty stemming from non-exact collocation between the aircraft in the Optimal Estimation retrieval. We added:

>  Uncertainties due to non-exact collocation between Polar 5 and 6 are neglected here. The average standard deviation of $Z_e$ is 0.7 dB over distances of 555 m, which corresponds to the mean horizontal distance between the aircraft, and therefore smaller than the assumed uncertainty of 1.5 dB. In Sect. 4 we discuss implications of the non-exact collocation on the presented results.

*2. P.7, l.171-175: I did not understand how the LWC values were estimated along the radar beam (in order to quantify the attenuation), this should be better explained.*

We apologize for keeping this analysis step confusingly brief. We extended:

> To estimate attenuation due to liquid water, we took LWC measurements from the Nevzorov probe operated onboard Polar 6 during the temporally closest vertical cloud profile. To obtain information on the vertical structure of clouds, Polar 6 flew vertical profiles in so-called "saw-tooth patterns". These patterns were flown in addition to straight legs at constant altitudes. Saw-tooth patterns are not well suited for good quality collocated measurements with Polar 5, where straight legs are preferred. Therefore a limited number of vertical profiles are available for each flight with collocation. During each flight analyzed in this study, at least three of such "saw-tooth patterns" were collected. Whenever Nevzorov probe measurements were not available, LWC was calculated by integrating the particle size distribution (PSD) of liquid particles (¡ 50 µm) measured with the cloud probes on board Polar 6. In both cases, LWC measurements were averaged to be on a regular vertical grid with a resolution of 10 m. Here, we neglect the distance traveled by Polar 6 during the profile, assuming LWC to be constant at each height bin. This assumption likely does not hold in reality, however, no measurements

with more precise information on horizontal and vertical LWC distributions are available.

*3. P.7, l.186: I think it should be "for" instead of "to" before "our results".*

Thank you.

*4. P.9, l.235: is $Z_e$ expressed in dBz or $mm^6 m^{-3}$?*

In dBZ, we added:

    ...at Polar 6 flight altitude in dBZ and...

*5. P.9, l.237: "to make $S_a$ more Gaussian": maybe showing a distribution (in appendix?) would strengthen the claim?*

We decided to remove this statement to avoid confusion. What we wanted to refer to was that the mass size parameter $a_m$ and $b_m$ from Maherndl et al. (2023) are a function of $M$ and that choosing $M$ in a logarithmic scale makes it possible to cover the whole range of mass size parameter (see Fig. 9 (a) from Maherndl et al. (2023)). However, since we only have one state space and one measurement parameter, $S_a$ is reduced to a scalar and and the original sentence does not make much sense.

*6. P.9, l.239: given that $S_y$ and $S_a$ are of the same magnitude, does the insensitivity to Sa imply that the 1st term in Eq(3) is dominant and hence that F() is strongly conditioning the retrieved values?*

Not really, because $S_y$ and $S_a$ have different units and are therefore not comparable. Additionally, our problem is well constrained and the cost function is expected to have one only minimum in most cases.

*7. P.11 l.286: it should be Fig4.b, no?*

Yes, thank you.

*8. P.13, l.307: it seems that "a" in between "point" and "perfect" should be removed.*

Done.

*9. P.13, l.312: the RMSE value seems much larger than the mean value, which suggests strong uncertainty no?*

We agree that it shows strong uncertainty in terms of temporal confluctuations, however the low ME shows good agreement in a statistical sense. We added:

    Mean error (ME) and root mean square error (RMSE) are <s>0.0077 and 0.032</s>

0.0026 and 0.031 for the point by point comparison. While we do not achieve good point by point agreement (large RMSE), both methods agree in a statistical sense (small ME).

*10. P.14, l.14: "simIlar"*

Done.

*11. P. 16, l.360: ' 'Figure 8 analyses the dependence of temperature and LWC on M '': should it be the other way around?*

Yes, we changed the sentence structure.

*12. P.18, l.411: I suggest to add "in terms of temporal cofluctuations" after "agreement" to clearly emphasize on what this agreement is.*

Good point, done.

*13. P.20, Fig.10: the dashed line in plots (b), (d)... is not explained in the caption.*

Thanks, we added the explanation in the caption.

*14. P.23, l.498: "depended": should it be "depending on the"?*

Yes, changed.

*15. P.23, l.513: the units of $N$ and $N_0$ should be $mm^{-3}mm^{-1}$, as $N(D)dD$ is the concentration of drops of size between $D$ and $D + dD$. This is consistent with the definition of $\Lambda$ 3 lines after (in its current version, $\Lambda$ would be dimensionless).*

Yes, thank you for catching the mistake. We changed:

> To approximate errors of the combined method $M$ retrieval, we present results obtained for synthetic data. We use the simulated rimed dendrite aggregates from Maherndl et al. (2023) binned into 10 logarithmic $M$ bins from $10^{-2}$ to $10^0$ ("true" $M$) and linear $D_{\max}$ bins from 0 to 10 mm with bin widths of 200 µm. We apply exponential PSDs $N(D) = N_0 \, \exp{(-\Lambda D)}$ to each $M$ bin, where $N$ is the number concentration in  $m^{-4}$ of particles of size $D$ in m, the intercept parameter $N_0$ (in  $m^{-4}$) describes the overall scaling and the slope parameter $\Lambda$ controls the shape.

---

## Author Response (AR2)

**Quantifying riming from airborne data during HALO-(AC)³**
**Response to the reviewers**

Nina Maherndl, Manuel Moser, Johannes Lucke, Mario Mech, Nils Risse, Imke Schirmacher, and Maximilian Maahn

January 23, 2024

*Original Referee comments are in italic*

> manuscript text is indented, with added text underlined and

We would like to thank the reviewers for their helpful comments. We revised the manuscript and responded to all of the reviewers' comments.

**1 Reviewer II**

**1.1 Summary**

*The authors have addressed my main concerns and the quality of the manuscript has improved. I list below a few typos and minor corrections to be taken care of (page and line numbers refer to the version with track changes):*

*1. p.7, l.166: "These functions are then applied to the 89 GHz TB". If "These functions" refers to retrieval mentioned just before, then I do not understand how it can be applied t Tb values... Please clarify.*

We apologize that the text was still confusing. We hope, we could clarify and refer to Ruiz-Donoso et al. (2020) for a more detailed explanation:

...the liquid water path (LWP) is estimated over open ocean only with a temporal resolution of 1 s as described in Ruiz-Donoso et al. (2020). The retrieval takes profiles of nearby dropsondes to calculate $T_B$ as a function of LWP measurements from simulations with the Passive and Active Microwave radiative TRAnsfer tool (PAMTRA, Mech et al., 2020).  $T_B$(LWP) is approximated by a third-order regression. The regression is then applied in an inverse scheme to the 89 GHz $T_B$ measurements to derive LWP...

*2. p.12, l.299: "the same range AS"*

Thank you, now fixed.

*3. p.14-15, l.362-364: I do not understand those numbers... And they do not sum up to 100%...*

The numbers add up to 100 % (and 99 % for the in situ method due to rounding) when adding the particle fractions with $M < 0.01$. Here, we want to show that most particles have $M$ between 0.01 and 0.1. The rimed fraction (= fraction of particles that are rimed) results depend on the threshold value we choose to separate "rimed" vs. "unrimed". To better clarify, we changed to:

> Assuming particles with $M < 0.01$ having negligible riming, we derive average rimed fractions of 88 % and 87 % over all collocated flight segments with the combined and the in situ method, respectively. These numbers appear quite high, however, they depend heavily on the rimed vs. unrimed threshold that is chosen; if we assume $M < 0.05$ to be unrimed instead of $M < 0.01$, we get 11 % and 9 % rimed particles, respectively. 12 % and 13 %  of particles have $M < 0.01$ for the combined and in situ method, respectively,  83 % and 83 % fall in range $0.01 \leq M < 0.1$, and only 5 % and 3 % have $M \geq 0.1$.

*4. p.16, l.414: "larger THAN zero"*

Thanks.

*5. p.26, l.609: I would use "homogeneous" rather than "continuous".*

Yes, we changed to "homogeneous".

*6. p. 28, l.664: should it be -10°C?*

Yes, thanks for catching that.

*7. As an additional point, I would like to mention that I am not sure I understand the response to my specific comment #6 (concerning Eq.3). The loss function defined in*

*Eq.3 is "mixing" different variables, so their respective magnitudes have an impact on the loss function value... And my point is not about a unique minimum or not, it is about the respective influence of the prior and the forward model.*

The different quantities in Eq. 3 are normalized by the covariance matrices $S_a$ and $S_y$. Selecting $S_a$ and $S_y$ determines whether more weight is put on the observations or the prior information. However, since we use only one observation variable to retrieve one state space variable this is not relevant for our study. For more in depth information about the Optimal Estimation theory see Rodgers (2000).